# Token-Scaled Logit Distillation for Ternary Weight Generative Language Models

**Minsoo Kim**[1]    **Sihwa Lee**[1]    **Janghwan Lee**[1]
**Sukjin Hong**[1,2]    **Du-Seong Chang**[2]    **Wonyong Sung**[3]    **Jungwook Choi**[1]*

[1]Hanyang University, Seoul, Republic of Korea
[2]KT, Seoul, Republic of Korea
[3]Seoul National University, Seoul, Republic of Korea
{minsoo2333, macto94, hwanii0288, choij}@hanyang.ac.kr
{sukjin.hong, dschang}@kt.com, wysung@snu.ac.kr

## Abstract

Generative Language Models (GLMs) have shown impressive performance in tasks such as text generation, understanding, and reasoning. However, the large model size poses challenges for practical deployment. To solve this problem, Quantization-Aware Training (QAT) has become increasingly popular. However, current QAT methods for generative models have resulted in a noticeable loss of accuracy. To counteract this issue, we propose a novel knowledge distillation method specifically designed for GLMs. Our method, called token-scaled logit distillation, prevents overfitting and provides superior learning from the teacher model and ground truth. This research marks the first evaluation of ternary weight quantization-aware training of large-scale GLMs with less than 1.0 degradation in perplexity and achieves enhanced accuracy in tasks like common-sense QA and arithmetic reasoning as well as natural language understanding.[2]

## 1   Introduction

Generative language models (GLMs) have made impressive strides in text generation, understanding, and reasoning, attracting significant attention in the field [1–7]. However, deploying GLMs remains a challenge due to their enormous model sizes. There is rising interest in practical GLMs with less than 10 billion parameters. Their capability can be improved through instruction fine-tuning [8–11]. For instance, Alpaca [12] showed that a fine-tuned 7 billion-parameter model can match the text generation performance of a 175 billion parameter GLM, highlighting the potential of smaller, more manageable models.

Since practical GLMs still contain billions of parameters, there is extensive research into model compression techniques for their efficient deployment. One such method is post-training quantization (PTQ), which simplifies the process by reducing bit-precision to 8 or 4 bits without the need for fine-tuning pre-trained GLMs [13–17]. This approach has gained traction due to its straightforward and fast processing time. However, it's been observed that these techniques cause a significant decrease in accuracy when the parameter count drops below 10 billion or when the bit-precision falls under 4 bits. As a result, there's a clear need for a more reliable quantization approach for GLMs with sub-4bit precision.

In response, we propose an alternative method, quantization-aware training (QAT), to address the issues PTQ poses for fine-tuned GLMs. QAT is a prevalent quantization technique that counteracts

---

*Corresponding Author
[2]Our code is available at https://github.com/aiha-lab/TSLD

37th Conference on Neural Information Processing Systems (NeurIPS 2023).

accuracy loss and attains a high compression rate for efficient deployment [18]. Notably, successful fine-tuning of sub-4bit natural language understanding models has been achieved through layer-to-layer (L2L) knowledge distillation (KD), a method used to offset errors resulting from aggressive quantization, such as binary or ternary weights [19–23]. However, applying QAT to GLM has limited success. While [24] introduced a token-level contrastive loss and [25] offered initial insights into the challenges of quantizing GLMs, both studies encountered substantial increases in perplexity in language modeling. Furthermore, no existing studies apply QAT to GLMs with billions of parameters, primarily due to the expensive nature of training with KD.

This study delves into the fundamental challenges of applying QAT to fine-tuned GLMs. We identify two main issues. First, the structure of a self-attention map in masked self-attention causes cumulative quantization errors across tokens, which conventional L2L KD struggles to compensate for. Second, the teacher-forcing mechanism [26] used in fine-tuning Transformer decoder necessitates ground-truth loss (GT Loss) – a factor largely overlooked in previous QAT methods – but including GT Loss risks overfitting. Our investigation reveals that logit distillation can overcome the limitations of L2L KD in token prediction recovery by reforming intermediate representations. Additionally, we found that applying token-wise logit scaling can significantly mitigate the risk of overfitting.

Drawing from our findings, we introduce a novel KD technique known as Token-Scaled Logit Distillation (TSLD), designed to enhance QAT for ternary quantization inference. We evaluate TSLD across a range of GLMs – originating from GPT-2 [2], OPT [4] and LLaMA [5] – of various sizes, including 7 billion models for the first time. The results show that TSLD achieves comparable, if not superior, performance in language modeling on ternary and 4-bit inference. When TSLD is applied to reasoning tasks, it surprisingly prevents overfitting to achieve task accuracy that is at least on par, if not better. These remarkable outcomes underline the potential of our proposed TSLD method in facilitating the deployment of ultra-low precision GLMs.

## 2 Related Work

**Fine-tuning for Generative Language Model.** GLMs are renowned for their unparalleled text generation, comprehension, and reasoning capabilities [1–7]. Studies reveal that their performance can be enhanced through instruction fine-tuning methods like Prefix-Tuning [11] or using natural language instructions and examples [8]. Fascinatingly, instruction-tuned models, including smaller ones, can outperform larger counterparts in specific tasks [9, 8, 12]. However, the vast parameter count in these models, compared to popular models such as BERT [27], can restrict their practical utility. To mitigate this, we suggest investigating efficient, lightweight techniques for GLMs encompassing up to 7 billion parameters.

**Quantization for Generative Language Model.** Quantization, a method that minimizes the inference cost of large models by utilizing a limited number of bits to represent weights, has been recently applied to GLMs [13–17]. This process has considerably cut down GPU memory usage and execution time. Two main types of quantization exist, quantization-aware training (QAT) and post-training quantization (PTQ), which differ in their requirement for re-training or fine-tuning. Although QAT has been effectively used in Transformer encoder models [22, 25], its application to GLMs poses challenges [24], with observed performance declines when applied to decoder-only models like GLMs [25]. This paper assesses the imbalance of quantization errors based on attention traits and language model generation patterns. We demonstrate that QAT can be conducted without substantial performance loss across various tasks, even in models exceeding one billion parameters.

**Knowledge Distillation for Language Model Compression.** Knowledge distillation (KD) is a prevalent transfer learning framework that imparts knowledge from a larger "teacher" model to a smaller "student" model, and it is effectively used to curb accuracy decline in models compressed through quantization-aware training (QAT) [28, 19, 21–23]. In encoder models, KD trains the quantized model (student) using the full-precision model's (teacher's) intermediate representation as the objective in QAT. Despite its effectiveness, this method requires more memory due to the intermediate representation from the teacher model and has been less explored in decoder models such as GLM [25, 24]. This paper introduces a novel, less memory-intensive KD method applicable to models with up to 7 billion parameters. We offer a thorough analysis of the teacher's information transfer in the decoder model and suggest a QAT-based KD method that retains minimal performance degradation, even when applying ternary weights to various GLMs.

## 3 Background and Challenges

### 3.1 Transformer Layer

Generative language models [1] are built with Transformer layers [29]. A standard Transformer layer includes two main sub-modules: Multi-Head Attention (MHA) and Feed-Forward Network (FFN). Input to the $l$-th Transformer layer is $\mathbf{X}_l \in \mathbb{R}^{n \times d}$ where $n$ and $d$ are the sequence length and hidden state size, respectively. Let $N_H$ be the number of attention heads and $d_h = d/N_H$. $\mathbf{W}_h^Q, \mathbf{W}_h^K, \mathbf{W}_h^V \in \mathbb{R}^{d \times d_h}$ are the weight parameters projecting $\mathbf{X}_l$ into Query ($\mathbf{Q}_h = \mathbf{X}_l \mathbf{W}_h^Q$), Key ($\mathbf{K}_h = \mathbf{X}_l \mathbf{W}_h^K$), and Value ($\mathbf{V}_h = \mathbf{X}_l \mathbf{W}_h^V$), respectively. The attention score ($\mathbf{A}_h$) is computed with the dot product of the projected Query and Key ($\mathbf{A}_h = \mathbf{Q}_h \mathbf{K}_h^\top$). The normalized version of this result is then passed through the softmax function and multiplied by the Value to get the output as $\text{head}_h = \text{softmax}(\mathbf{A}_h / \sqrt{d_h}) \mathbf{V}_h$. Then, the output of the Multi-Head Attention (MHA) is defined as follows:

$$\text{MHA}(\mathbf{X}_l) = \text{Concat}(\text{head}_1, ..., \text{head}_{N_H}) \mathbf{W}^O. \tag{1}$$

FFN consists of two fully-connected layers with weight parameters $\mathbf{W}^1$ and $\mathbf{W}^2$:

$$\text{FFN}(\mathbf{Y}_l) = \text{GeLU}(\mathbf{Y}_l \mathbf{W}^1 + b^1) \mathbf{W}^2 + b^2. \tag{2}$$

Therefore, the operations at the $l$-th Transformer layer can be defined as:

$$\mathbf{Y}_l = \mathbf{X}_l + \text{MHA}(\text{LayerNorm}(\mathbf{X}_l)); \quad \mathbf{X}_{l+1} = \mathbf{Y}_l + \text{FFN}(\text{LayerNorm}(\mathbf{Y}_l)). \tag{3}$$

### 3.2 QAT with KD for Transformer Decoders

QAT emulates inference-time quantization during training to learn parameters robust to the quantization error. In particular, ternary quantization represents all the weight parameters ($\mathbf{W}^Q, \mathbf{W}^K, \mathbf{W}^V, \mathbf{W}^O, \mathbf{W}^1, \mathbf{W}^2$) into ternary values $\mathbf{t} \in \{+1, 0, -1\}^k$ along with a scale factor $\alpha$ for sub-2bit inference at deployment. In this work, we follow the approach of TWN [30] that analytically estimates the optimal $\alpha$ and $\mathbf{t}$ to minimize $\|\mathbf{w} - \alpha \mathbf{t}\|_2^2$, where $\mathbf{w} = \text{vec}(\mathbf{W})$ and $k$ is the number of elements of the weight parameters.

Due to aggressive bit-reduction, ternary quantization causes significant accuracy loss. KD can help compensate for accuracy degradation, where the original full-precision model works as a teacher to guide the training of the quantized model as a student. In case of Transformer models, prior works [19, 21, 22, 31] applied KD on every Transformer layer's output activation $\mathbf{X}_l$ as well as attention scores $\mathbf{A}_l$ with mean squared error (MSE) loss, denoted as $L_{L2L}$:

$$L_{L2L} = \sum_{l=1}^{L+1} \text{MSE}(\mathbf{X}_l^S, \mathbf{X}_l^T) + \sum_{l=1}^{L} \text{MSE}(\mathbf{A}_l^S, \mathbf{A}_l^T), \tag{4}$$

where superscripts $S$ and $T$ represent the student and teacher models, respectively.

The final output logits of the student ($\mathbf{Z}^S$) and the teacher ($\mathbf{Z}^T$) are used to compute the cross-entropy loss. Given that $N$ is the total number of the tokens in input sequences, and $V$ is the vocabulary size the language model can recognize and generate. Using the softmax function, we can convert each model's $i_{th}$ token prediction logit output into probability distributions, which are then utilized for the loss for logit distillation($L_{logit}$):

$$\mathbf{P}_i = \frac{e^{\mathbf{Z}_{i,j}}}{\sum_{j=1}^V e^{\mathbf{Z}_{i,j}}}, \quad L_{logit} = -\frac{1}{N} \sum_{n=1}^N \sum_{i=1}^V \mathbf{P}_{n,i}^T \log(\mathbf{P}_{n,i}^S). \tag{5}$$

The overall loss for KD is generally computed as $L_{KD} = L_{L2L} + L_{logit}$, without GT Loss as noted in previous studies [19, 22, 31]. Yet, some methods utilize only $L_{logit}$ [24]. Our study underscores the necessity of integrating $L_{logit}$ and the GT Loss for an effective application of QAT in Transformer decoders.

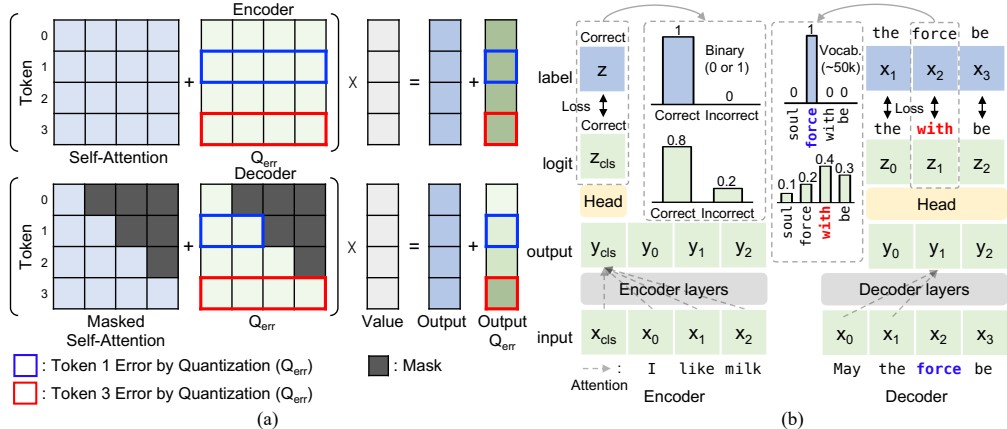

Figure 1: (a) Illustration of attention mechanism in the encoder (top) and decoder models (bottom). (b) Left: performing NLU task [32] by encoder model. Right: performing language modeling task by decoder model with teacher-forcing (input token is independent of the previously generated token)

## 3.3 Quantization Challenges on GLMs

In this section, we compare the computations of Transformer encoders and decoders to deepen our understanding of the fresh challenges that surface within the realm of GLMs.

**Cumulative Quantization Errors in Causal Attention.** Causal attention, which integrates masking into self-attention to avoid referencing future tokens, is vital for causal language modeling tasks. To comprehend the quantization characteristics of GLMs, we contrast the computation procedures of the self-attention map. For a Transformer encoder, the quantization error reflected on the self-attention map due to the process of projecting Query, Key, and Value is evenly spread across tokens because of the token-parallel nature of computing inherent in Transformer encoders. However, the mask in the causal attention accumulates quantization errors from each token, creating uneven errors in the output computation.

Fig. 1(a) contrasts the attention mechanism of the encoder and decoder model under quantization. The encoder's self-attention is illustrated at the top of the figure, decomposed into self-attention and quantization error ($Q_{err}$) components. The combined attention probabilities are utilized in a weighted sum with the Value, where tokens 1 and 3 (in a blue and red box respectively) are affected by an identical number of attention probabilities with quantization error. Conversely, the decoder's causal attention, shown at the bottom, uses only the attention probabilities of the current token and its preceding ones. For instance, the Value for token 1 in the bottom of Fig. 1(a) (in a blue box) uses only two attention probabilities affected by quantization error, while token 3 (in a red box) includes those from all preceding tokens. This illustration highlights that causal attention inherently leads to a disproportionate accumulation of quantization errors in the latter tokens. Thus, we need a decoder QAT strategy that addresses this imbalance within the causal attention module. In Section 4.1, we assess the limitations of current KD methods in managing cumulative quantization errors and introduce an enhanced logit distillation error compensation mechanism.

**Necessity of Ground Truth Loss.** In fine-tuning, encoder models, often used in Natural Language Understanding (NLU), and decoder models, common in Natural Language Generation (NLG), employ distinct mechanisms for receiving GT Loss, as shown in Fig.1(b). Encoder models for NLU tasks use a single special token to compute cross-entropy loss with a limited number of classes [27], as depicted on the left of Fig.1(b). On the other hand, decoder models in NLG tasks predict each subsequent token, transforming each token's representation into a logit vector with a class size equivalent to the vocabulary size, often exceeding 50k [1], shown on the right of Fig. 1(b). This process allows decoder models to obtain GT Loss for each input token, providing detailed token-level prediction information. Given these differences, there is a compelling need to consider the necessity of GT Loss in the decoder model's QAT in a token-wise manner. However, previous QAT [24] on the decoder models neglects the consideration of GT Loss due to a perceived degradation in performance when GT Loss is utilized. Accordingly, Section 4.2 offers an in-depth analysis of the interplay between KD and GT Loss during the QAT.

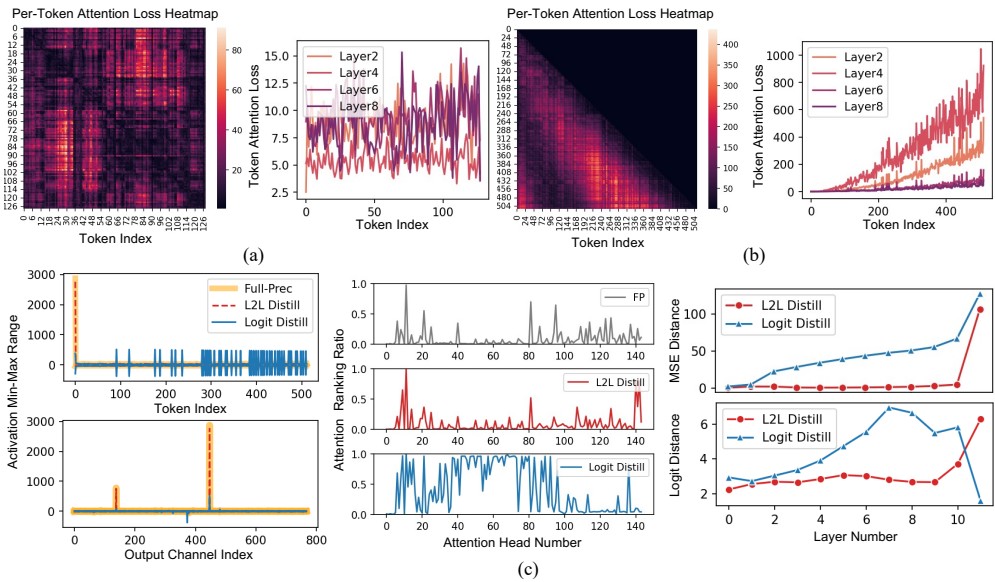

Figure 2: Comparison of weight quantization error on attention map through MSE loss in (a) encoder (BERT-base, RTE task) and (b) decoder (GPT-2, PTB task) model. (c) Left: min-max dynamic range per layer (Logit Distill vs L2L Distill). Middle: attention ranking ratio comparison (FP vs Logit Distill vs L2L Distill). Right: per layer token-wise logit distance and MSE distance

# 4 Method

## 4.1 Logit Distillation for Cumulative Quantization Error

**Motivation.** The inherent nature of causal attention, where each token representation builds upon the representation of the previous tokens, presents previously unseen challenges when applying quantization to decoder models. For a clearer understanding of the decoder model, we conduct a comparative analysis with the encoder model to examine the impact of quantization error on the model. In Fig. 2 (a), the quantization error of the encoder self-attention map exhibits a widespread presence of errors due to the absence of masking in self-attention, and the per-token quantization errors along the layers also show irregular patterns depending on the token index. However, in Fig. 2 (b), the heat map of the decoder model reveals an increasing brightness of quantization errors as we move toward the later tokens. When examining the token index, the phenomenon of quantization errors accumulating toward the later tokens becomes even more pronounced. This previously unconsidered phenomenon of token quantization error accumulation in the decoder model is a crucial feature to consider in GLM QAT. Reflecting on this feature, we analyze the effectiveness of prior KD methods for language modeling and explore suitable KD approaches for the decoder model. Analysis on cumulative quantization error for a wider variety of GLMs can be found in Appendix A.3.

**Comparison of KD Methods for Decoder QAT.** Building on a deeper comprehension of the decoder model, we evaluate the efficiency of current KD methods for QAT in decoders and propose an enhanced KD approach informed by our decoder model analysis. We analyze how two different KD methods, Layer-to-Layer distillation (L2L KD) and logit distillation (Logit KD), tackle systematic outliers in QAT [15], using the min-max dynamic range per token and per channel of each layer's intermediate output. As shown in Fig. 2(c) left, both KD methods demonstrate distinct strategies in addressing the teacher model's systematic outliers. While L2L distillation guides the QAT process to mirror the outliers of the teacher model, Logit KD deviates from this pattern, generating new outliers not seen in the teacher model. These outliers consistently emerge in specific channel indices where the teacher model's outliers are present. Additionally, to compare the relative token attending order within each QAT model's self-attention map, we employ a ranking ratio comparison method [22]. This technique conveys the average relative importance of a single token within each attention map. As depicted in Fig. 2(c) middle, the L2L KD method closely mirrors the teacher model's ranking changes. However, the Logit KD method exhibits substantial variation in this ranking shift within a certain head range.

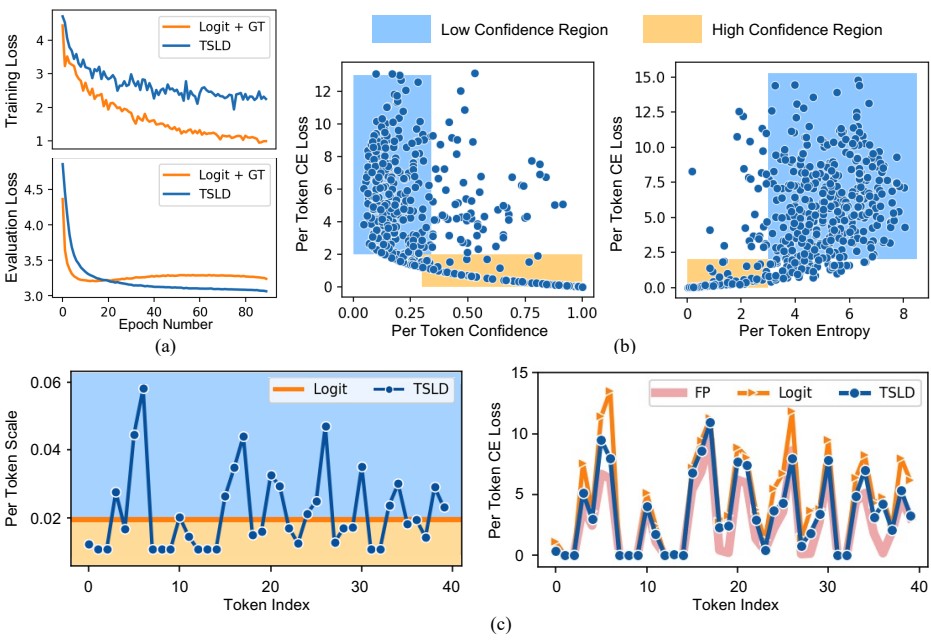

Figure 3: (a) Training/evaluation loss curve with different QAT-KD methods. (b) token-wise prediction statistics scatter plot (left: cross entropy loss and token confidence, right: cross entropy loss and token entropy). (c) Impact of TSLD: per-token scale and cross-entropy loss (left: per-token scale, Right: per-token cross-entropy loss). Analysis utilizes OPT-125m for PTB task language modeling. See Appendix A.2 for further analysis on other GLMs

**Logit Distillation for Token-wise Prediction Recovery.** We further analyze the QAT model's token logit distributions. Since token representations evolve along the layers to form the next token's probability [33], we assess each layer's logit distribution and the logit distance from the teacher model. As depicted in Fig. 2(c), L2L KD creates a token representation that closely mirrors the teacher model in both logit distribution and mean-squared error (MSE) distance during mid-layer stages but fails to match the final logit distribution. Conversely, Logit KD, despite diverging from the teacher model's logit distribution in the middle layers, accurately reproduces the final logit distribution. These observations highlight Logit KD's distinct mechanism for token-wise prediction recovery, managing cumulative quantization error in decoder models. In intermediate layers, Logit KD varies the attention values across channels as shown in Fig.2(c), leading to a diverging token representation from the FP model, with this middle stage adjustment acting to counteract accumulated quantization errors in later tokens. Consequently, Logit KD aligns the final logit distribution for each token, crucial for the accuracy of causal language modeling. Therefore, Logit KD, aligning with the characteristics of the decoder model, stands out as a natural choice for QAT. The subsequent section will delve into previously unexamined issues encountered by Logit KD in decoder QAT.

## 4.2 Token-Scaled Logit Distillation for Avoiding Overfitting with GT Loss

**Motivation.** This section tackles the overfitting problem arising from the combination of Logit KD and GT Loss during QAT. We also investigate the probabilistic behavior displayed by the decoder model in language modeling tasks. The study by [24] highlights instances where employing GT Loss and Logit KD adversely impacts the performance of decoder QAT. To understand this issue better, we conduct tests using Logit KD both independently and combine with ground truth loss. As depicted in Fig. 3 (a), overfitting is observed in the QAT when both ground truth loss and Logit KD are applied.

**Understanding Causes of Overfitting.** To better understand the causes of overfitting, we analyze the logit output for each token that the teacher model generates during language modeling. From the logit information by the teacher model, we derive the probability distribution ($\mathbf{P}_i^T = \text{softmax}(\mathbf{Z}_i^T)$) for $i_{th}$ token prediction. Based on this distribution, we further calculate cross-entropy ($-y_{n,i} \log(\mathbf{P}_{n,i}^T)$), confidence score ($\max(\mathbf{P}_i^T)$) and entropy ($-\sum_i \mathbf{P}_i^T \log(\mathbf{P}_i^T)$) for each token prediction. These

metrics unveil a confidence disparity in language modeling—a trend uniformly observed across decoder models of varying scales. The cross-entropy loss of token prediction logits plotted against the probability confidence score, as illustrated in Fig. 3 (b), distinctly demarcates the *Low Confidence Region* (blue box) with low probability confidence and high cross-entropy loss from the *High Confidence Region* (yellow box) with high probability confidence and low cross-entropy loss. This observation implies a potential overlap between high-confidence Logit KD and the role of GT Loss cross-entropy, suggesting that redundant information from high-confidence Logit KD might mirror the effects of ground truth loss, thereby contributing to the overfitting observed.

**Token-Scaled Logit Distillation (TSLD)**. Based on investigations into the probabilistic relation of token predictions and overfitting in QAT, we propose an adaptive KD method that adjusts Logit KD based on token confidence. This approach utilizes the phenomenon of confidence disparity in token predictions from the teacher model. Our method, called Token-Scaled Logit Distillation (TSLD), de-emphasizes Logit KD for high-confidence tokens to prevent overfitting in QAT while emphasizing Logit KD for low-confidence tokens possessing with a high entropy probability distributions. Specifically, low-scaled Logit KD (high confidence, low entropy) effectively reduces the overlap with GT Loss, leading to an improvement in overfitting. On the other hand, high-scaled logit KD (low confidence, high entropy) emphasizes the distillation of more informative token prediction distribution from the teacher model, which has rich soft label information.

$$CE_n^T = -\sum_{i=1}^{V} y_{n,i} \log(\mathbf{P}_{n,i}^T), \quad \text{scale}_n = \frac{e^{CE_n^T/\tau}}{\sum_{k=1}^{N} e^{CE_k^T/\tau}} \tag{6}$$

$$L_{TSLD} = \sum_{n=1}^{N} \left( \text{scale}_n \times -\sum_{i=1}^{V} \mathbf{P}_{n,i}^T \log(\mathbf{P}_{n,i}^S) \right) \tag{7}$$

The implementation of TSLD is straightforward. By considering the relationship between token confidence and token prediction cross entropy loss in Fig 3(b), we can determine the $n_{th}$ token scale values ($\text{scale}_n$) based on the cross entropy loss of the teacher model ($CE_n^T$) using softmax function as shown in Eq. 6. Note that $y_{n,i}$ is a ground truth label where $y_{n,i} = 1$ if token $i$ is the true next token at position $n$ and $\tau$ is the temperature parameter for softmax function. The scale of the each token ($\text{scale}_n$) is then applied in the logit distillation by multiplying it with the cross entropy loss between the student and teacher models as shown in Eq. 7. As depicted in Fig. 3(c) left, the scale values for token-specific Logit KD are determined adaptively based on the per-token cross-entropy loss of the teacher model. In the right graph of Fig. 3(c), we can observe that tokens with higher cross entropy loss in the teacher model correspond to higher scale values in the per token scale graph, compared to Logit KD method applying the same scale value ($1/N$) across all tokens as shown in the left graph of Fig. 3(c).

TSLD brings about two significant effects by applying different scales based on confidence disparity, with negligible computational overhead. As shown in Fig. 3(a), TSLD de-emphasizes Logit KD for high-confidence tokens, thereby preventing overfitting. Conversely, for low-confidence tokens possessing a high entropy probability distribution, TSLD emphasizes the Logit KD. This action allows the student model to more closely mimic the teacher model's cross-entropy loss as seen in the right graph of Fig. 3(c). A detailed analysis of the TSLD method's computational cost can be found in Appendix A.1.

## 5 Experiments

### 5.1 Experimental Settings

In this section, we evaluate the effectiveness of TSLD in the QAT of various sizes of decoder models with sub-4bit quantization. We've set up comparative experiments to demonstrate the proficiency of our TSLD method against existing PTQ and other QAT KD methods. Our findings illustrate that TSLD substantially enhances both the language modeling performance (measured by Perplexity or PPL) and the accuracy in tasks related to reasoning (common-sense QA and arithmetic) and natural language understanding.

**Task and Models**. We evaluate our proposed method for language modeling (PTB [34]), common-sense QA tasks (PIQA [35], OpenbookQA [36], ARC_easy [37], ARC_challenge [37] and arithmetic

| Precision | Quantization Method | Optimization Method | GPT-2 | | | | OPT | | | |
|---|---|---|---|---|---|---|---|---|---|---|
| | | | 0.1B | 0.3B | 0.8B | 1.5B | 0.1B | 1.3B | 2.7B | 6.7B |
| FP16 baseline | | | 20.91 | 18.21 | 15.20 | 14.26 | 18.17 | 13.75 | 11.43 | 10.21 |
| W4A16 | PTQ | OPTQ [14] | 22.41 | 19.35 | 17.26 | 15.86 | 19.75 | 14.30 | 11.82 | 11.73 |
| | QAT | Logit [24] | 20.98 | 18.54 | 16.79 | 15.42 | 17.60 | **13.73** | 11.82 | 11.20 |
| | | Logit+GT | 21.51 | 18.58 | 15.49 | 14.89 | 19.63 | 15.03 | 12.58 | 11.78 |
| | | TSLD | **19.95** | **17.53** | **15.32** | **14.50** | **17.45** | 13.90 | **11.59** | **11.00** |
| W2A16 | QAT | L2L+Logit [25] | 23.79 | 21.21 | 17.80 | 15.82 | 20.47 | 17.62 | 14.67 | 11.75 |
| | | Logit [24] | 22.84 | 19.87 | 16.46 | 15.27 | 18.86 | 14.80 | 12.26 | 11.33 |
| | | Logit+GT | 23.80 | 20.20 | 17.77 | 16.52 | 21.62 | 16.41 | 13.20 | 12.41 |
| | | TSLD | **21.74** | **18.57** | **16.14** | **15.02** | **18.58** | **14.60** | **11.97** | **11.17** |

Table 1: Perplexity comparison in GPT-2 and OPT series across various model sizes (0.1B to 6.7B) on the PTB dataset with QAT-KD (tensor-wise) and PTQ (channel-wise) quantization methods

| QAT KD Method | PIQA | | OpenbookQA | | ARC_easy | | ARC_challenge | | GSM8K | |
|---|---|---|---|---|---|---|---|---|---|---|
| | ACC (↑) | PPL (↓) | ACC (↑) | PPL (↓) | ACC (↑) | PPL (↓) | ACC (↑) | PPL (↓) | ACC (↑) | PPL (↓) |
| OPT-2.7B FP16 | 76.71 | 10.91 | 49.60 | 26.16 | 66.12 | 7.41 | 37.20 | 8.96 | 20.39 | 2.07 |
| Logit [24] | 74.32 | 11.69 | 45.40 | 29.41 | 58.92 | 9.05 | 31.91 | 12.38 | 20.02 | **2.03** |
| GT+Logit | 74.97 | 12.10 | 46.20 | 31.08 | 58.84 | 8.66 | 32.16 | 12.04 | 19.56 | 2.12 |
| TSLD | **75.62** | **11.35** | **46.81** | **28.93** | **59.39** | **8.12** | **33.45** | **11.05** | **20.24** | 2.03 |

| Model | GPT-Neo-1.3B | OPT-6.7B | | LLaMA-7B | | |
|---|---|---|---|---|---|---|
| QAT KD | PTB (PPL) | GSM8K (ACC/PPL) | | PTB (PPL) | GSM8K (ACC/PPL) | |
| FP16 | 17.62 (↓) | 22.52 (↑) | 1.89 (↓) | 8.76 (↓) | 30.25 (↑) | 1.47 (↓) |
| Logit [24] | 21.01 | 21.08 | **1.93** | 12.22 | 25.47 | **1.52** |
| TSLD | **19.27** | **24.49** | 2.14 | **11.60** | **26.23** | **1.52** |

Table 2: Top: Results for the OPT-2.7B model fine-tuned on common-sense QA and arithmetic reasoning task using various QAT-KD (tensor-wise) methods. Bottom: QAT-KD (channel-wise) results on language modeling task and arithmetic reasoning task across various GLM models.

reasoning based text-generation task (GSM8K [38]). Additionally, our assessment extends to Natural Language Understanding (NLU) task (GLUE [39]), ensuring a comprehensive analysis. Our benchmark models encompass widely used GLMs, such as GPT-2 [2], OPT [4], GPT-Neo [40] and LLaMA [5] [41] with various sizes ranging from 0.1B to 7B parameters.

**Fine-Tuning Settings**. In fine-tuning the language modeling task, we employ a chunk-based pre-processing method: all training datasets are concatenated, then split into shorter chunks defined by input sequence length. For reasoning task fine-tuning, we utilize a sentence-based approach, concatenating each dataset's question and answer parts to form new sentences, individually used as the fine-tuning dataset. Detailed hyper-parameter settings and other specifics are in Appendix C.2. Experiments are conducted on an A100-40GB GPU. Our QAT experiments start with models that have undergone task-specific fine-tuning. During quantization, the KD process employs the FP fine-tuned model as the teacher model, while the quantized model acts as the student.

**Implementation Settings**. We devise a QAT-KD framework that leverages pipeline parallelism with PyTorch Pipe API enabling the training of models with capacities exceeding 1.3 billion. We apply weight quantization to the matrix multiplication layers in each decoder layer of the GLM. We conduct experiments on L2L KD [25] but encounter out-of-memory problems for models with more than 1.3B parameters in A100-40GB GPU. This issue is believed to arise due to the requirement for both the teacher and student models to store the outputs generated by all of their respective intermediate layers during the knowledge distillation process. GPU memory consumption comparisons for each QAT-KD method can be found in the Appendix A1.

## 5.2 Evaluation on Language Modeling Task

Table 1 outlines the performance comparison of TSLD with leading PTQ and QAT methods [14, 24] for language modeling of PTB dataset. For 4-bit weight quantization, OPTQ sees a notable performance drop in GPT-2 and OPT models up to 6.7 billion parameters, aligning with the original paper's observations [14]. However, QAT methods show lower perplexity due to weight parameters fine-tuned for robust reduced-precision inference. QuantGPT [24], which exclusively uses Logit KD achieves impressive perplexity, whereas Logit+GT KD sees degradation. Conversely, TSLD offers

| Precision | QAT KD Method | CoLA | | MRPC | | SST-2 | | RTE | |
|---|---|---|---|---|---|---|---|---|---|
| | | ACC ($\uparrow$) | PPL ($\downarrow$) | ACC ($\uparrow$) | PPL ($\downarrow$) | ACC ($\uparrow$) | PPL ($\downarrow$) | ACC ($\uparrow$) | PPL ($\downarrow$) |
| OPT-1.3B FP16 | | 61.03 | 1.34 | 81.92 | 2.58 | 94.26 | 2.00 | 76.53 | 3.94 |
| W4A16 | OPTQ [14] | **54.61** | **1.36** | **80.14** | **2.43** | **95.07** | **2.02** | **56.32** | **3.96** |
| | AWQ [13] | 13.63 | 1.45 | 66.42 | 3.49 | 94.26 | 2.02 | 54.51 | 4.72 |
| | Logit [24] | 50.76 $_{\pm2.35}$ | 1.36 | 81.94 $_{\pm1.48}$ | 2.62 | 93.57 $_{\pm0.23}$ | 2.09 | 75.23 $_{\pm0.83}$ | 4.34 |
| | GT+Logit | 54.07 $_{\pm0.34}$ | **1.34** | 83.17 $_{\pm0.51}$ | 2.60 | 93.34 $_{\pm0.22}$ | 2.11 | 75.31 $_{\pm1.07}$ | 4.09 |
| | TSLD | **56.33** $_{\pm0.98}$ | **1.34** | **83.33** $_{\pm1.22}$ | **2.52** | **94.05** $_{\pm0.19}$ | **2.04** | **75.97** $_{\pm0.31}$ | **4.05** |
| W2A16 | Logit [24] | 48.72 $_{\pm2.68}$ | 1.37 | 81.62 $_{\pm0.62}$ | 2.79 | 93.08 $_{\pm0.35}$ | 2.11 | 74.15 $_{\pm1.36}$ | 4.72 |
| | GT+Logit | 50.10 $_{\pm1.38}$ | **1.34** | 82.10 $_{\pm0.99}$ | 2.65 | 92.77 $_{\pm0.28}$ | 2.14 | 73.79 $_{\pm1.16}$ | 4.44 |
| | TSLD | **54.47** $_{\pm1.47}$ | **1.34** | **82.20** $_{\pm0.94}$ | **2.63** | **93.92** $_{\pm0.29}$ | **2.06** | **75.31** $_{\pm0.54}$ | **4.36** |

Table 3: Results for the OPT-1.3B model fine-tuned on GLUE [39] using different QAT-KD methods with five random seed tests. Channel-wise quantization is applied in both PTQ and QAT-KD.

the lowest perplexity, underlining token-wise scaling method's effectiveness in incorporating GT knowledge. Remarkably, TSLD's performance boost allows QAT models to match full-precision performance across various capacity ranges in all decoder models.

For 2-bit weight quantization, L2L KD sees significant accuracy degradation, and Logit+GT KD suffers from overfitting. TSLD outperforms Logit KD [24] across all model sizes, maintaining PPL degradation of no more than 1.0 from the baseline. We also tested the general applicability of TLSD on popular open-sourced GLM models (GPT-Neo-1.3B [40], LLaMA-7B [5]) in language modeling PTB task. Table 2-below indicates that TSLD consistently surpassed the competitor, Logit KD [24]. Notably, 2-bit TSLD utilizes simple ternary weight quantization, which is hardware-friendly.

## 5.3 Evaluation on Reasoning Task

We assess the effectiveness of our proposed method in commonsense QA (PIQA, OpenbookQA, ARC_easy, ARC_challenge) and reasoning-based text-generation task (GSM8K) employing the LM Evaluation Harness framework from EleutherAI [42]. Given the capacity requirements for reasoning tasks, we use OPT-2.7B/6.7B and LLaMA-7B models as baselines, rather than smaller models. Table 2 presents a performance comparison of 2-bit weight quantization with different KD methods. In commonsense QA tasks, TSLD consistently showcased the lowest perplexity and, consequently, the highest accuracy, drawing the 2-bit quantization results even closer to FP performance as shown in Table 2-top.

Considering the GSM8K task, Table 2 reveals that TSLD outperformed Logit KD in terms of perplexity and accuracy with OPT-2.7B/6.7B and LLaMA models. Notably, while QuantGPT (Logit KD) achieves comparable or better perplexity, its reasoning task accuracy is lower, potentially due to insufficient GT information. Conversely, TSLD achieves excellent reasoning accuracy while maintaining competitive perplexity, underscoring TSLD's ability to balance language modeling and reasoning performance through its token-wise scaling to avoid overfitting. The generated text sample results of the GSM8K task are provided in the Appendix D.

## 5.4 Evaluation on Language Understanding Task

We fine-tune the decoder model for Natural Language Understanding (NLU) tasks using a language modeling approach as illustrated in Fig. 1(b). In our experiments outlined in Table 3, we compare the performance of the latest PTQ methods (AWQ [13], OPTQ [14]) and QAT-KD methods with OPT-1.3B model. For 4-bit quantization, the PTQ technique shows a noticeable degradation in performance compared to QAT results, excluding SST-2 task. TSLD achieves the lowest perplexity and the highest accuracy across all the experiments except SST-2, where its accuracy is in-par with the full-precision case. These findings demonstrate that TSLD can robustify the performance of 4-bit quantized GLMs for various NLU tasks, while 4-bit PTQ may suffer from performance degradation.

In ternary quantization, TSLD consistently outperforms the alternative QAT-KD methods for all the cases, demonstrating its superior performance in bridging the accuracy gap with the full-precision cases. Interestingly, ternary TSLD even achieved similar or superior accuracy compared to 4-bit PTQ in many tasks (e.g., CoLA, MRPC, SST-2), highlighting its benefits on both accuracy and memory savings.

| Model config. | 6.7B | | | 13B | | | 175B | | |
|---|---|---|---|---|---|---|---|---|---|
| Input channel | 4096 | 4096 | 16384 | 5120 | 5120 | 20480 | 12288 | 12288 | 49152 |
| Output channel | 4096 | 16384 | 4096 | 5120 | 20480 | 5120 | 12288 | 49152 | 12288 |
| FP32 baseline | 0.067 | 0.201 | 0.194 | 0.100 | 0.285 | 0.287 | 0.391 | 1.472 | 1.522 |
| 8-bit | 0.039 | 0.068 | 0.068 | 0.043 | 0.095 | 0.092 | 0.121 | 0.407 | 0.395 |
| ($\times$ speedup) | $\times$1.71 | $\times$2.93 | $\times$2.83 | $\times$2.32 | $\times$3.01 | $\times$3.13 | $\times$3.23 | $\times$3.61 | $\times$3.85 |
| 4-bit | 0.030 | 0.057 | 0.057 | 0.041 | 0.076 | 0.075 | 0.096 | 0.326 | 0.318 |
| ($\times$ speedup) | $\times$2.23 | $\times$3.52 | $\times$3.40 | $\times$2.45 | $\times$3.75 | $\times$3.82 | $\times$4.07 | $\times$4.51 | $\times$4.78 |
| 2-bit | 0.025 | 0.053 | 0.053 | 0.039 | 0.072 | 0.064 | 0.077 | 0.221 | 0.232 |
| ($\times$ speedup) | $\times$2.68 | $\times$3.77 | $\times$3.65 | $\times$2.57 | $\times$3.95 | $\times$4.48 | $\times$5.07 | $\times$6.66 | $\times$6.56 |

Table 4: Kernel execution time (msec)

### 5.5 Ablation Study

**Reduced-Precision Kernels Execution Time.** We developed custom CUDA kernels to enhance inference speed with applied (2-,4-,8-bit) quantization. Like OPTQ, we packed the weights to minimize the model size and load overhead. Our kernel eliminates the need for weight unpacking during the model forward pass, resulting in a speedup shown in Table 4. We tested our kernel mainly on models larger than 6.7B, where weight load overhead is notably high. The reported times are the average execution time for 10,000 kernel runs on a single A100-80GB GPU. For the FP32 baseline, we used PyTorch's nn.Linear. As shown in Table 4, our 2-bit kernel for the 175B model can potentially accelerate a single matrix multiplication operation by an average of approximately 6.1 times compared to FP32.

| Precision | Granularity | GPT-2 | | | OPT | | GPT-Neo | LLaMA |
|---|---|---|---|---|---|---|---|---|
| | | 0.1B | 0.3B | 0.8B | 0.1B | 1.3B | 1.3B | 7B |
| FP16 baseline | | 20.91 | 18.21 | 15.20 | 18.17 | 13.75 | 17.62 | 8.76 |
| W2A16 | Tensor-wise | 21.74 | 18.57 | 16.14 | 18.58 | 14.60 | 30.60 | 12.31 |
| | Channel-wise | **21.30** | **18.48** | **15.97** | **18.42** | **14.46** | **19.27** | **11.60** |

Table 5: Comparison of tensor-wise and channel-wise quantization across various GLMs (GPT-2, OPT, GPT-Neo and LLaMA). The TSLD KD method is employed in this experiments

**Quantization Granularity Impact.** To account for output channel weight variations, channel-wise quantization is used [15, 14]. By integrating our QAT-KD approach with channel-wise quantization, we can achieve further performance enhancement. An interesting observation emerges: the gains from channel-wise quantization vary by the type of GLM. As illustrated in Table 5, for the GPT-2 and OPT series, the PPL performance increase due to channel-wise quantization is less than 1. However, for GPT-Neo and LLaMA, the performance enhancement effect resulting from channel-wise quantization is significantly pronounced. This variation in performance gains suggests distinct channel-wise weight distributions across different GLM models. A detailed analysis of the weight distribution for each GLM is addressed in the Appendix A.4.

## 6 Conclusion

We introduce token-scaled logit distillation, a new approach for Quantization-Aware Training of Generative Language Models. This method effectively reduces overfitting and enhances learning from the teacher model and ground truth. Importantly, this research is the first to evaluate ternary quantization-aware training on large-scale GLMs, achieving less than 1.0 perplexity degradation and preserving commonsense QA and arithmetic reasoning task accuracy.

## Acknowledgments and Disclosure of Funding

This work was supported by Institute of Information & Communications Technology Planning & Evaluation (IITP) grangts funded by the Korea government (MSIT) (2020-0-01373, Artificial Intelligence Graduate School Program Hanyang University, 2023-RS-2023-00253914, artificial intelligence semiconductor support program to nurture the best talents), the National Research Foundation of Korea (NRF) grant, funded by the Korea government (MSIT) (No. 2021R1A2C1013513, No. RS-2023-00260527), and the Artificial Intelligence Industrial Convergence Cluster Development Project, funded by the Ministry of Science and ICT (MSIT, Korea) and Gwangju Metropolitan City.

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

# A    Supplementary Analysis

## A.1    Computation Requirements of TSLD

| Training Method | QAT-KD Method | GPT2-0.3B 512 | | GPT2-0.8B 512 | | OPT-1.3B 256 | | OPT-1.3B 1024 | |
|---|---|---|---|---|---|---|---|---|---|
| | | Speed (iter/sec) | Memory (MiB) | Speed (iter/sec) | Memory (MiB) | Speed (iter/sec) | Memory (MiB) | Speed (iter/sec) | Memory (MiB) |
| QAT | GT | 2.03 | 19663 | 1.03 | 36317 | 5.15 | 10051 | 2.83 | 15167 |
| QAT-KD | Logit | 1.57 | 22622 | 0.81 | 40989 | 4.44 | 11589 | 2.27 | 17529 |
| | GT+Logit | 1.56 | 22622 | 0.81 | 40989 | 4.44 | 11589 | 2.27 | 17529 |
| | L2L+Logit | 1.51 | 31462 | **OOM** | **OOM** | 4.28 | 12143 | 2.12 | 25315 |
| | TSLD | **1.57** | **22622** | **0.81** | **40989** | **4.43** | **11589** | **2.26** | **17529** |

Table A1: QAT memory consumption and training speed study for KD method. The results are reported on the PTB dataset on the Ternary QAT-KD of GPT-2 and OPT series models with input sequence lengths ranging from 256 to 1024

The TSLD method integrates token-wise cross-entropy loss with Logit KD, involving two operations as detailed in Eq.7. Specifically, the term $\sum_{i=1}^{V} y_{n,i} \log(P_{n,i}^T)$ computes the cross-entropy loss from teacher logits($Z_n^T$). This result, processed through a softmax function, derives the scaling value for each token. Multiplied element-wise with Logit KD term, $\sum_{i=1}^{V} P_{n,i}^T \log(P_{n,i}^S)$, it yields a token-wise scaled Logit KD. In fact, TSLD leverages the teacher logits that are pre-computed in Logit KD, circumventing extra memory usage. Furthermore, the associated computations have a complexity of $O(N)$, making TSLD's overhead negligible for training.

To evaluate TSLD's efficiency, we detail training speeds and GPU memory consumption for various QAT-KD methods using GPT-2 models in Table A1 left. Compared to Logit-based methods, TSLD maintains speed without extra memory consumption. In contrast, L2L KD stores intermediate activations from both the teacher and student models for KD, resulting in significantly increased memory requirements, evident from Table A1 left. As model size grows, as evidenced in scenarios utilizing GPT2-Large, memory requirements rise, leading to "Out of Memory" errors on an A100-40GB GPU. These findings highlight efficacy of TSLD, enhancing QAT-KD performance with memory comparable to Logit KD, while L2L KD demands significantly more. Even when the sequence length is extended from 256 to 1024, as Table A1 right shows,TSLD maintains the same GPU memory consumption and training speed as the Logit KD method.

## A.2    Token Confidence Disparity Analysis

Our analysis of confidence disparity in token predictions, detailed in Section 4.2, extends beyond a specific GLM model. In fact, this observed trend is consistently present across various GLM models. As shown in Fig A1, we can distinctly observe the emergence of the *Low Confidence Region* (blue box) and the *High Confidence Region* (yellow box) consistently across models: OPT-6.7B(left), LLaMA-7B(middle), and LLaMA-2-7B(right). Additionally, as shown in Fig A1 right, we plot the token prediction's statistics with varying input sequence lengths of 128 and 512. Regardless of the sequence length, the demarcation of confidence disparity remains consistent. This observation demonstrates that the TSLD methodology, grounded in the the probabilistic dynamics of token prediction, can be universally applied across various GLMs.

## A.3    Cumulative Quantization Error Analysis with LLM

In this section, we aim to expand our analysis of the cumulative quantization error discussed in Section 4.1 to GLMs larger than 6B parameters. By implementing 2-bit ternary quantization [30] on the OPT-6.7B and LLaMA-7B models, we assess the attention map quantization error in comparison to the FP model through MSE loss. These errors are visualized using a heatmap plot (Fig. A2 top), and the average attention map loss per token was plotted against each layer (Fig. A2 below). For the OPT-6.7B model, quantization error is measured for the 5th and 15th layers. Regarding the LLaMA-7B model, quantization errors are depicted for input sequence lengths of 128 and 512.

For the OPT-6.7B model at its 5th layer and the LLaMA-7B model with a sequence length of 128, we note an accumulation of quantization errors towards the latter tokens, as discussed in Section 4.1.

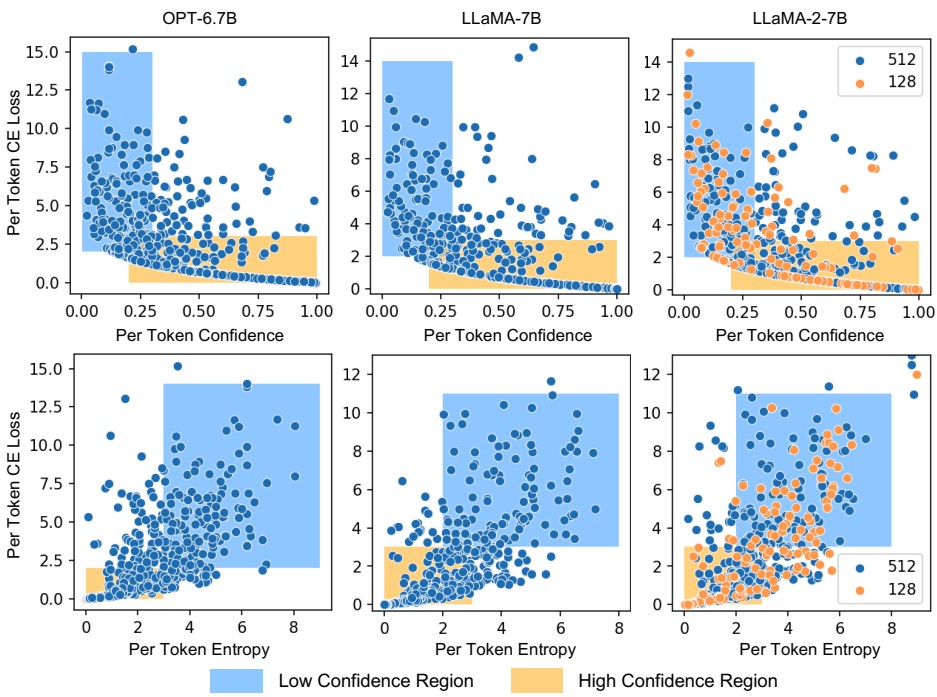

Figure A1: Scatter plots representing probabilistic relations of token prediction. The top plots show CE loss versus confidence for each token prediction, while the bottom plots plot CE loss with entropy. From left to right: OPT-6.7B, LLaMA-7B, and LLaMA-2-7B. For LLaMA-2, results for two input sequence lengths (128, 512) are plotted. Input dataset is wikitext-2

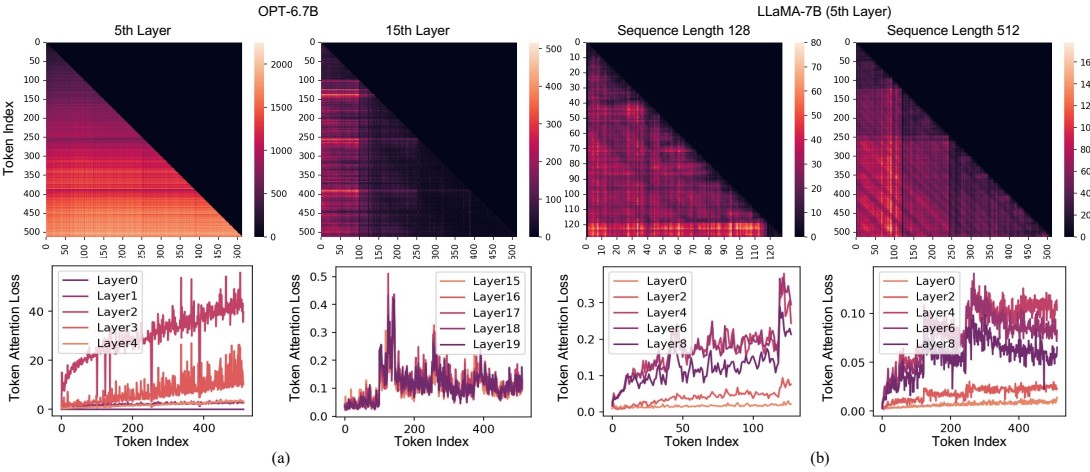

Figure A2: Top: Heat map of 2-bit ternary weight quantization error on attention map MSE loss. (a) OPT-6.7B's 5th and 15th layer attention loss. (b) LLaMA-7B attention loss for sequence lengths 128 to 512. Below: Average per-token attention MSE loss across layers from each attention map loss heatmap

However, as we delve deeper into the layers of OPT-6.7B or introduce longer input sequences to LLaMA-7B, this phenomenon becomes less pronounced. We speculate that this attenuation might arise from the intricate interplay of quantization errors as the depth of GLM increases, and the evolving attention patterns associated with varying sequence lengths influencing accumulation of quantization errors. A thorough exploration of cumulative quantization errors for larger GLMs will be reserved for future research.

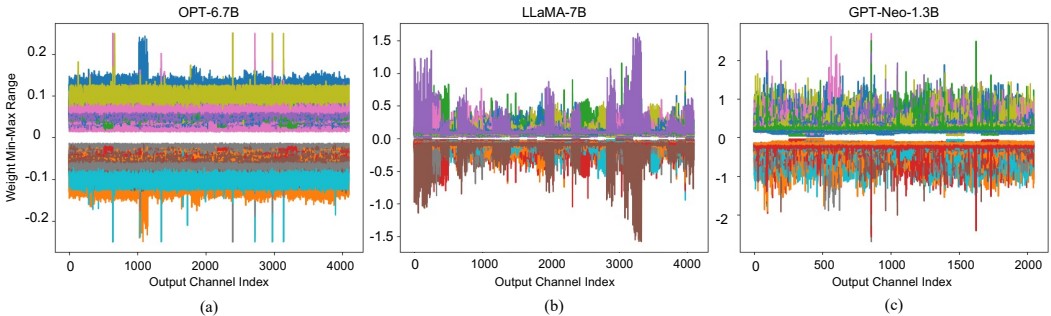

Figure A3: Min-Max range of Linear module weights for three types of GLMs per output channel. In the graphs, each color represents the min-max range of an each module weight. (a) OPT-6.7B (b) LLaMA-7B (c) GPT-Neo-1.3B.

## A.4 Comparison of GLM Weight Distribution

Pre-trained GLMs show a wide variety of weight distributions [43]. We examine the Min-Max range of weights for each linear module across the output channel in various GLM models (OPT, LLaMA, and GPT-Neo) as visualized in Fig. A3. This analysis aims to elucidate the performance disparities observed in Section 5.5 due to quantization granularity (tensor-wise and channel-wise). For the OPT model, we observe that each module exhibits a consistent channel-wise min-max range, which is notably narrow, spanning from -0.2 to 0.2. In contrast, both LLaMA and GPT-Neo showcase a much more diverse min-max range across output-channels for each module, with the range itself being significantly broader, approximately from -2 to 2. This diversity in the output channel-specific min-max range clarifies the performance differences between tensor-wise and channel-wise approaches, as highlighted in Table 5. Specifically, OPT, which has limited output channel diversity, showed minimal performance differences between tensor-wise and channel-wise methods. Conversely, models like GPT-Neo and LLaMA, characterized by extensive channel diversity, exhibit significantly enhanced performance with channel-wise quantization. These findings suggest that determining the appropriate quantization granularity in QAT, with the aim of minimizing quantization error, necessitates a comprehensive understanding of the channel-wise weight distribution of the target GLM.

# B Supplementary Experimental Results

## B.1 8-bit Activation Quantization

Table A2 showcases experimental results applying both ternary weight quantization and 8-bit activation quantization (W2A8). We apply min-max quantization for activation quantization in the same way as in [19] [24] [25] [22], taking into account the asymmetric distribution of certain activation parts. Specifically, asymmetric min-max quantization is implemented in the multiplication of the Query and Key in self-attention and in the input activation of the FC2 linear layer [3].

In W2A8, in line with observations from Section 5.2, L2L KD exhibits substantial accuracy degradation than Logit KD. Although Logit + GT performs less optimally than Logit KD due to the previously mentioned overfitting impact, our method outperforms the others across all model sizes, thereby underscoring the effectiveness of the TSLD method.

## B.2 Clipping Value Exploration in 4-bit Weight Quantization

When adopting a QAT method like QuantGPT, which determines the clipping value with a learnable scale factor, it's crucial to initialize the scale factor appropriately to match the weight distribution of the quantized model. In the case of the OPT model, a much narrower distribution is observed compared to GPT-2, as illustrated in Fig. A4(a). If we set the clipping value ($\gamma$=1.0) in the same way as QuantGPT, we can observe that over 40% of weight elements are detrimentally clipped, as shown in Fig. A4(b). To alleviate the destructive clipping phenomenon in 4-bit quantization, we conduct an

---

[3]We use activation quantization code in the following repository https://github.com/huawei-noah/Pretrained-Language-Model/tree/master/TernaryBERT

| Precision | Optimization Method | GPT | | | | OPT | | | |
|---|---|---|---|---|---|---|---|---|---|
| | | 0.1B | 0.3B | 0.6B | 1.5B | 0.1B | 1.3B | 2.7B | 6.7B |
| FP32 baseline | | 20.91 | 18.21 | 15.20 | 14.26 | 18.17 | 13.75 | 11.43 | 10.21 |
| W2A8 | L2L+Logit[25] | 24.88 | 21.61 | - | - | 20.50 | - | - | - |
| | Logit [24]. | 23.14 | 20.13 | 16.59 | 15.34 | 19.21 | 15.28 | 12.87 | 11.70 |
| | Logit+GT. | 24.37 | 20.78 | 18.01 | 16.87 | 21.59 | 16.58 | 13.49 | 12.81 |
| | TSLD | **22.01** | **18.83** | **16.26** | **15.23** | **18.92** | **14.95** | **12.14** | **11.43** |

Table A2: Impact of activation quantization (Ternary weight, 8-bit Activation Quantization results) in QAT-KD (tensor-wise)

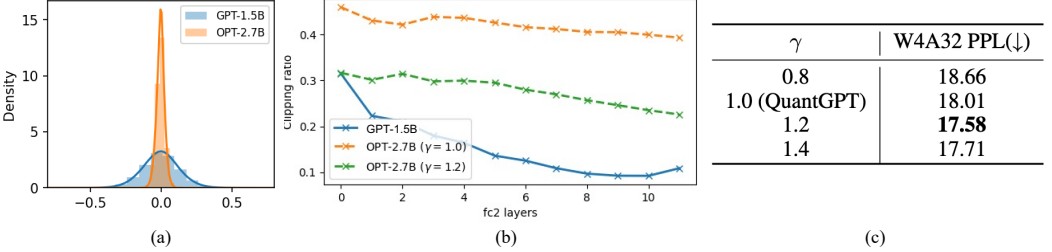

Figure A4: (a) Weight distribution of GPT-2-1.5B/OPT-2.7B (4th layer, FC-1) (b) We measure the ratio representing how many weight elements were clipped in the FFN-2 layers weight quantization. Upon applying QAT with the original QuantGPT recipe ($\gamma = 1.0$), we observe that over 40% of values were clipped in OPT-2.7B, a significantly higher rate compared to GPT-2-1.5B. (c) $\gamma$ initialization exploration PPL results in OPT-0.1B with PTB dataset.

experiment exploring the initial value of the $\gamma$ scale in QuantGPT. ($\gamma$ scale determines clipping value in QuantGPT. Detailed quantization implementations of QuantGPT are further elaborated in C.2) Through exploration of $\gamma$ initialization, we are able to reduce the proportion of clipped weights as in Fig. A4(b) by increasing the initial value of the $\gamma$ scale, and consequently, achieving performance improvement as shown in Fig. A4(c). Through exploring $\gamma$ scale hyper-parameters tailored to the OPT weight distribution, we manage to fairly compare multiple KD methodologies in 4-bit OPT QAT without the adverse effects of excessive quantization clipping. These experimental results suggest that, when initializing the clip value in the learnable clipping QAT method, one should consider the weight distribution characteristics of the target GLM.

## B.3 Results of Decoder-Style BERT QAT

In Section 4.1, we discuss the cumulative quantization error due to the structural feature of the GLM's masked self-attention, and compare the effectiveness of Logit KD and L2L KD in the QAT. In this experiment, we compare distillation methods in the Encoder model (BERT-base [27]), where, due to the absence of masking, the quantization error is evenly distributed among all tokens. According to [22], L2L KD is crucial in the Encoder model QAT KD, and having more layers to distill has proven beneficial for QAT performance.

As explained in Section 3.3, in the Natural Language Understanding tasks of the encoder model, we calculate the cross entropy loss using the representation of a single special token (class token, [CLS]) as logits. Drawing from the fact that the decoder model's language modeling fine-tuning uses cross entropy loss of all token representations, we attempt to use every token's final representation outputs as logits in the encoder model and measure cross entropy loss with the teacher model's final token representation logits and use this loss as Logit KD (we call this KD method "Logit - All Token").

| Task | RTE | STS-B | MRPC | CoLA |
|---|---|---|---|---|
| Full precision | 73.28 | 89.24 | 87.77 | 58.04 |
| Logit - [CLS] Token (Logit KD) | 55.59 | 86.46 | 82.43 | 38.60 |
| Logit - All Tokens | 70.54 | 87.46 | 87.03 | 48.36 |
| Logit - [CLS] Token + L2L (L2L KD) | **72.34** | **88.98** | **87.70** | **51.12** |

Table A3: QAT-KD (tensor-wise) performance with multiple KD options on selected GLUE [39] tasks with BERT-base [27] model.

As can be seen in Table A3, the Logit - All Token method, which utilizes all final token representations as logits, is considerably more beneficial for performance than utilizing a single special token's representation as Logit (Logit - [CLS] Token). However, when compared with Logit - [CLS] Token + L2L KD, we found that employing L2L KD yields superior performance in the QAT of the encoder model.

This additional experiment reveals that in an encoder model QAT KD where the quantization error is distributed among all tokens, L2L KD, which forces the student model's each layer output to closely mimic that of the teacher model, is the most effective distillation method in QAT. This understanding extends our comprehension of how to adjust QAT KD methodologies to accommodate the structural nature of each model.

## C   Experimental Details

### C.1   Model Description

| Configuration | GPT | | | | OPT | | | | GPT-Neo | LLaMA |
|---|---|---|---|---|---|---|---|---|---|---|
| | 0.1B | 0.3B | 0.6B | 1.5B | 0.1B | 1.3B | 2.7B | 6.7B | 1.3B | 7B |
| # of Layers | 12 | 24 | 36 | 48 | 12 | 24 | 32 | 32 | 24 | 32 |
| # of Hidden Dim | 768 | 1024 | 1280 | 1600 | 768 | 2048 | 2560 | 4096 | 2048 | 4096 |
| # of Head | 12 | 16 | 20 | 25 | 12 | 32 | 32 | 32 | 16 | 32 |
| Learning Rate (FP) | 1e-4 | 1e-4 | 1e-4 | 1e-4 | 1e-4 | 1e-4 | 5e-5 | 5e-5 | 1e-4 | 5e-5 |
| Epoch (FP) | 3 | 3 | 3 | 3 | 3 | 3 | 3 | 3 | 3 | 1 |
| Learning Rate (QAT) | 1e-4 | 1e-4 | 1e-4 | 1e-4 | 1e-4 | 1e-4 | 1e-4 | 5e-5 | 1e-4 | 7e-5 |
| Epoch (QAT) | 90 | 60 | 30 | 30 | 90 | 30 | 30 | 10 | 30 | 5 |

Table A4: Configuration of each pre-trained decoder model with various sizes and hyper-parameter selection for fine-tuning FP and QAT-KD. All experiments consistently set a batch size of 4, and sequence length of 512 in language modeling fine-tuning

In our experiments, we conduct task specific fine-tuning for various pre-trained GLMs (GPT-2 [2], OPT [4]), GPT-Neo [40], and LLaMA [5]) at various sizes (0.1B to 7B). The GPT-2 pre-trained model has a vocabulary size ($V$) of 50257 and employs the GeLU activation function [44]. The OPT pre-trained model features a vocabulary size ($V$) of 50272 and uses the ReLU activation function [45]. On the other hand, the GPT-Neo pre-trained model has the same vocabulary size ($V$) as OPT and utilizes the new GeLU activation function [46]. It also incorporates Rotary Positional Embedding (RoPE) [47] for positional embeddings. As for the LLaMA pre-trained models, they have a vocabulary size ($V$) of 32000 and utilize the SwiGLU activation function [48]. These models also employ RoPE for positional embeddings. For detailed configuration information for each model size, please refer to Table A4.

### C.2   Quantization Details

**Quantization Aware Training with KD.** In order to use KD in QAT, we need to initialize teacher and student models respectively. The teacher model undergoes task-specific fine-tuning in full precision (FP) based on a pre-trained model. The student model is then initialized from the teacher model, after which quantization is applied. The hyper-parameter settings of FP fine-tuning and QAT-KD, across various model types and sizes, can be found in Table A4. Furthermore, our experimental implementation utilizes the Huggingface language modeling code base[4].

**Post Training Quantization.** We conduct our experiments of post training quantization with OPTQ and AWQ [14, 13], using the code from original paper respectively [5] [6]. We utilize a calibration dataset comprising 128 randomly selected 2048 token segments from the PTB [34] dataset for OPTQ and Pile [49] dataset for AWQ. To ensure a fair comparison with QAT, we adopt per-channel quantization as our quantization granularity.

---

[4]https://github.com/huggingface/transformers/tree/main/examples/pytorch/language-modeling
[5]https://github.com/IST-DASLab/gptq
[6]https://github.com/mit-han-lab/llm-awq

**QuantGPT Implementation.** In this paper, we primarily draw comparisons with QuantGPT, a state-of-the-art methodology above prior works regarding decoder QAT. This approach introduces two main contributions: a module-dependent scaling method and token-level contrastive distillation.

For 4-bit QAT-KD experiments, we adopt the QuantGPT [24] quantization method (module-dependent dynamic scaling). QuantGPT considers the quantization scale factor as a learnable parameter and optimizes it through QAT. Following the dynamic scaling method of QuantGPT, we determine the clipping value $\alpha$ for quantization by multiplying the average weight magnitude $\frac{\|\mathbf{w}\|_1}{n}$ with a learnable scale factor $\gamma$, where $\|\cdot\|_1$ denotes $\ell_1$ norm: $\alpha = \gamma \cdot \frac{\|\mathbf{w}\|_1}{n}$. In this case, the initial value for the $\gamma$ is set to 1, and the learning rate for $\gamma$ is 0.0002.

Upon implementing token-level contrastive distillation, we observe issues of robustness in replicating the token-level contrastive distillation KD method, where incorrect choices in negative sampling could lead to performance degradation [7]. Therefore, To ensure a fair comparison, we exclude the contrastive loss from our implementation of Logit KD.

**ALPACA-style Fine-Tuning for Arithmetic Reasoning Task.** In arithmetic reasoning task (GSM8K) fine-tuning, We employ the ALPACA style fine-tuning method [12], proposed for instruction-following demonstration fine-tuning. This fine-tuning method fundamentally employs a language modeling approach, as demonstrated in Fig. 1(b), predicting the next word in a sequence. However, the ALPACA-style fine-tuning process has a distinctive characteristic: it transforms the datasets into a format that comprises instruction-response pairs, as illustrated in Table A5. We apply this ALPACA-style fine-tuning method to large pre-trained GLMs exceeding 2 billion parameters (OPT-2.7B/6.7B, LLaMA-7B).

# D   Examples of Arithmetic Reasoning Text Generation

In this section, we examine the QAT KD method on arithmetic reasoning task through a comparison of generation results from the QAT model. The GSM8K dataset serves as a benchmark for measuring arithmetic reasoning abilities, and models are expected to generate text responses auto-regressively based on the questions provided. This task requires not only correct mathematical calculations to produce the right answer, but also a logical problem-solving process, and the final answer is correct if both the logic and calculations are accurate. In evaluating GSM8K, we employed a greedy decoding strategy for the text generation process.

In Table A5, we can observe that the answers generated by the QuantGPT QAT model appear to make sense at first glance (corresponding to low PPL results in Table 2), but upon closer examination, it becomes evident that the necessary problem-solving process and computations are incorrect. Particularly in Question 1, the model writes that it should perform multiplication in the solution process, but actually executes division, leading to an incorrect intermediate result. From there, it continues to develop an entirely wrong solution. In Question 2, while the solution process and calculations align, incorrect methods are used to derive the intermediate results, eventually leading to a wrong answer. In Table A6, it can be seen that in Question 3, the model skips necessary intermediate steps in the problem-solving process, resulting in an incorrect answer. In Question 4, while the solution process is correct, there are errors in the calculation leading to an incorrect answer. In contrast, we can confirm that the generation results using our method accurately list the problem-solving process necessary to solve the question, just like the ground truth. Moreover, the calculation results are accurate, thereby resulting in the correct answer.

Such sample comparisons allow us to see the limitations of using Logit KD alone in reasoning tasks. As shown in Table A5 and A6, when Logit KD is used solely for QAT, as the low PPL result in Table 2 shows, it appears to well recover the natural text generation ability of the teacher model. However, in the results of the reasoning tasks that require a logical progression of solutions and accurate calculations, deficiencies such as errors in the solution process and calculation errors are observed. With our method that effectively combines GT Loss and Logit KD, we achieve the natural text generation capabilities of the teacher model while simultaneously enhancing logical reasoning and accurate calculation abilities through the ground truth during the QAT. Consequently, our method exhibits superior accuracy in the GSM8K benchmark.

---

[7]This issue has been acknowledged in the revised QuantGPT paper [50]

**Question 1:** *Below is an instruction that describes a task, paired with an input that provides further context. Write a response that appropriately completes the request.* The zookeeper feeds all the apes in the zoo. He orders all the bananas from a local farm every 2 months. If the monkeys need 200 bananas, the gorillas need 400 bananas, and the baboons need 100 bananas every month, how many bananas does he need to order to last for 2 months? *Response:*

**Ground Truth:**

The total number of bananas the prime apes need every month is 200+400+100= «200+400+100=700»700.

Therefore in 2 months, the total number of bananas they need is 700x2= «700*2=1400»1400. #### 1400

**QuantGPT:**

The zoo needs 200 bananas/monkey x 2 monkeys = «200*2=400»400 bananas for the gorillas.

It needs 400 bananas/gorilla x 4 gorillas = «400*4=1600»1600 bananas for the baboons.

It needs 1600 bananas/baboon x 100 baboons = «1600/100=16»16 bananas for the baboons.

It needs 16 bananas/baboon x 4 baboons = «16*4=64»64 bananas for the baboons.

It needs 64 bananas/baboon x 4 baboons = «64*4=256»256 bananas for the zoo.

It needs 256 bananas/2 months = «256/2=128»128 bananas to last for 2 months. #### 128

**Ours:**

The zoo needs to buy 200 bananas/month x 2 months = «200*2=400»400 bananas.

The gorillas need 400 bananas/month x 2 months = «400*2=800»800 bananas.

The baboons need 100 bananas/month x 2 months = «100*2=200»200 bananas.

The total number of bananas needed is 400 bananas + 800 bananas + 200 bananas =

«400+800+200=1400»1400 bananas. #### 1400

**Question 2:** *Below is an instruction that describes a task, paired with an input that provides further context. Write a response that appropriately completes the request.* The employees of Google went on a day trip. 4 buses were rented that have the capacity of holding 60 employees. 6 minibusses that can hold 30 employees, and 10 minivans that can hold 15 employees. How many employees can join the day trip? *Response:*

**Ground Truth:**

60 x 4 = «60*4=240»240 employees can ride on the bus.

30 x 6 = «30*6=180»180 employees can ride on the minibusses.

15 x 10 = «15*10=150»150 employees can ride on the minivans.

Therefore 240 + 180 + 150 = «240+180+150=570»570 employees can join the day trip. #### 570

**QuantGPT:**

The capacity of each bus is 60 employees / 4 buses = «60/4=15»15 employees.

The capacity of each minibus is 30 employees / 6 minibus = «30/6=5»5 employees.

The capacity of each minivan is 15 employees / 10 minivans = «15/10=1.5»1.5 employees.

The total capacity of the buses and minibus is 15 employees + 5 employees + 1.5 employees = «15+5+1.5=20»20 employees.

The total number of employees who can join the day trip is 20 employees + 60 employees = «20+60=80»80 employees. #### 80

**Ours:**

4 buses can hold 60 x 4 = «60*4=240»240 employees.

6 minibusses can hold 30 x 6 = «30*6=180»180 employees.

10 minivans can hold 15 x 10 = «15*10=150»150 employees.

Therefore, 240 + 180 + 150 = «240+180+150=570»570 employees can join the day trip. #### 570

Table A5: Samples of arithmetic reasoning outputs generated by OPT-6.7B ternary weight quantized with different methods. *Italics* part refers to the instruction formatting component in question.

**Question 3:** *Below is an instruction that describes a task, paired with an input that provides further context. Write a response that appropriately completes the request.* On a particular week, a tow truck pulled ten cars for each of the first three days and then four fewer cars on each of the remaining days of the week. Calculate the total number of cars it towed that week. *Response:*

**Ground Truth:**

On the first three days, towing ten cars a day, the tow truck pulled 3*10 = «3*10=30»30 cars.

It pulled four fewer cars each day, which is 10-4 = «10-4=6»6 cars on the remaining days.

If it pulled ten cars for three days and four less on the remaining days, then it pulled for 7-3 = 4 days, four cars less each day.

For the four days, the car pulled 6*4= «6*4=24»24 cars.

The total number of cars it pulled that week is 24+30 = «24+30=54»54 cars.

#### 54

**QuantGPT:**

On the first three days, the truck towed a total of 3*10 = «3*10=30»30 cars.

On the remaining days of the week, it towed a total of 10-4 = «10-4=6»6 cars.

The total number of cars it towed that week is 30+6 = «30+6=36»36 cars.

#### 36

**Ours:**

On the first three days, the truck pulled 10*3 = «10*3=30»30 cars.

On the remaining days, it pulled 10-4 = «10-4=6»6 cars each day.

The total number of cars it pulled on the remaining days is 6*4 = «6*4=24»24

In total, the truck pulled 30+24 = «30+24=54»54 cars that week.

#### 54

**Question 4:** *Below is an instruction that describes a task, paired with an input that provides further context. Write a response that appropriately completes the request.* Nick, Richard, Jason and DJ each have paintball guns. DJ has 8 guns, Nick has 10 guns, RJ has 1 gun and Richard has 5 guns. If they were to share their guns equally, how many guns would each of them have? *Response:*

**Ground Truth:**

The total number of guns is 8+10+1+5 =«8+10+1+5=24» 24 guns.

They would each have 24/4=«24/4=6» 6 guns.

#### 6

**QuantGPT:**

DJ has 8 guns, Nick has 10 guns, RJ has 1 gun and Richard has 5 guns.

For a total of 8+10+1+5 = «8+10+1+5=22» 22 guns.

If they were to share their guns equally, each of them would have 22/4 = «22/4=5» 5 guns.

#### 5

**Ours:**

DJ has 8 guns, Nick has 10 guns, RJ has 1 gun Richard has 5 guns.

Total they have 8+10+1+5 = «8+10+1+5=24» 24 guns.

If they were to share them equally, each of them would get 24/4 = «24/4=6» 6 guns

#### 6

Table A6: Samples of arithmetic reasoning outputs generated by OPT-6.7B ternary weight quantized with different methods. *Italics* part refers to the instruction formatting component in question.

