# OpenReview forum: "Token-Scaled Logit Distillation for Ternary Weight Generative Language Models"
_NeurIPS.cc/2023/Conference — NeurIPS 2023 poster_

### Official Review · Reviewer_cozw · 2023-06-12

**Soundness:** 3 good
**Presentation:** 3 good
**Contribution:** 3 good
**Rating:** 5
**Confidence:** 4

**Summary:**

The work proposes modifications for Quantization-Aware Training (QAT) that utilize knowledge distillation and takes into account token-scaled logit distillation for preventing overfitting. They show the motivation for their method through a deep analysis and present the empirical merits of this method via language modeling and mathematical reasoning tasks.

**Strengths:**

The proposed Token-Scaled Logit Distillation (TSLD) method as well as the analysis that comes before  is novel and interesting. I like the fact that the authors detected a problem with standard KD and proposed a nice way to mitigate it which replaces the uniform weighting over the tokens. I’m not sure though about the time complexity it takes to compute this loss, as well as the quadratic nature of this complexity given larger sequences, due to the nested summation. I’d suggest discussing this in a future revision.

Generally the paper is well written and motivated. The authors present a series of analysis that showcase the importance of utilizing their method during quantized training.

**Weaknesses:**

Minor comments:
* L17: state that ALPACA was fine-tuned from the LLaMA 7B model.
* The notations in Section 3.2 do not follow standard mathematical rules (e.g., matrix is denoted by two letters, a vector letter  is capitalized, etc.) and prior works. Please check previous works describing the transformer modules and change the equations accordingly.
* L119: Masked self-attention is the wrong wording here. You probably meant causal attention.
* Section 3.3 is written in a vague form and too informally. Consider adding a mathematical proof for the error accumulation in causal models.

Major comments
* What about long-range transformer models? How will the suggested method influence them?
* In L50 you state one example for a task (arithmetic reasoning). What about other tasks? There is a missing analysis/discussion why would this task specifically benefit from the proposed method, while other tasks don’t.
* I know that the paper was submitted before the QLORA was published, but in case this paper is accepted, I would expect the authors to add a comparison or at least a discussion about the merits of the proposed method against QLORA and other quantized low-rank adapter techniques. I do not consider this point in the final score calculation of this paper.
* An inaccurate point in Section 3.3 second part (Necessity of Ground Truth Loss) is the fact that decoder-only models use the entire vocabulary for prediction, unlike encoder models that use only the number of classes. This is also not correct, since one may consider and optimize a few tokens instead of the entire vocabulary, and thus there is a slight difference between encoders and decoders in that case for classification tasks.
* Empirical results over two datasets is not enough for me, and I wonder how the performance will change given more language modeling and other NLU tasks.
* There is an empirical evaluation regarding inference speed, but I wonder how fast is training time. This is crucial, since pre-training for example is costly anyway.

Missing references
* Jin, Jing, et al. "KDLSQ-BERT: A quantized bert combining knowledge distillation with learned step size quantization." arXiv preprint arXiv:2101.05938 (2021).

**Questions:**

Please respond to the concerns stated under the Weaknesses section.

**Limitations:**

I don't see any limitations section, please review and add this. One limitation can be for example the extra training time required by this method.

---

> ### Author Rebuttal · Authors · 2023-08-10
>
> Thank you to the reviewer for your valuable and constructive comments. To address the queries posed, our explanations are provided below.
>
> **Comment 1) Expanding TSLD Evaluation Across NLU/CSQA Tasks
> (Major Comments 2,5)**
>
> **Answer 1)** To validate the effectiveness of TSLD, we broadened our comparative experiments to encompass four CSQA datasets and additional NLU datasets, as suggested by the reviewer. As displayed in Tables R1 and R2, TSLD consistently exhibited the lowest perplexity and therefore the highest accuracy. This success is most pronounced in NLU's CoLA and RTE tasks, shown in Table R1, where TSLD notably outperformed traditional QAT-KD methods in terms of accuracy. Moreover, in all four CSQA tasks outlined in Table R2, TSLD's performance was superior, bringing the 2-bit quantization results closer to full precision (FP) performance. These experimental results further underscore the effectiveness of our TSLD approach.
>
> **Comment 2) Computational complexity of TSLD: Training time of long range Transformers (Strength, Major comments 1 and 6)**
>
> **Answer 2)** Thank you for pointing out these essential aspects of computational complexity and the application to long-range transformers. We'll address the training time cost of our proposed method as follows:
>
> - **Training time of QAT with KD for Transformer decoders (Table R7):** QAT's tuning of weight parameters is costly, especially with larger model sizes. However, TSLD does not incur additional overhead compared to QAT-KD. Specifically, the previous KD method, L2L-KD, necessitate additional memory requirements to store intermediate activation data. However, our proposed TSLD requires only the teacher's last logit, avoiding this memory overhead. As seen in Table R7, this memory savings enables QAT for larger models, unlike L2L, which struggles with out-of-memory problems for models of 0.8B or larger.
>
> - **Computational complexity of TSLD wit long sequence (Table R7,R8):** TSLD's computational complexity consists of two main operations in addition to basic Logit-KD, both of O(N) complexity, so it doesn't increase quadratically with larger sequences. This results in almost no additional training speed relative to standard Logit-KD (Table R7). We also considered the reviewer's focus on long-range transformers, examining training costs related to input sequence lengths. Table R8 shows the training costs for various QAT KD methods as sequence length varies. Remarkably, TSLD's complexity only modestly increases with sequence length, causing a negligible rise in training time without extra memory use.
>
>
> **Comment 3) Comparison with QLoRA (Major comment 3)**
>
> **Answer 3)** Thank you for bringing attention to interesting aspects. QLoRA enhances the accessibility of LLM by enabling parameter-efficient fine-tuning, even at lower precision levels. However, its approach differs fundamentally from QAT in that it doesn't directly incorporate the quantization error during training; i.e., it requires FP16 adaptors to be processed separately during inference, causing non-trivial latency overhead. To this end, merging QLoRA with QAT to conduct QAT in a more parameter-efficient manner would present an intriguing direction for future work.
>
>
> **Comment 4) Discussion about class sizes between Transformer encoders and decoders. (Major comment 4)**
>
> **Answer 4)** Thank you for highlighting these details. We concur that the number of classes can change depending on the task, but our main emphasis was on the distinction in how encoder and decoder models typically handle GT loss. While encoders, aiming for sequence comprehension, use a limited number of tokens to compute GT loss, decoders, working on causal language modeling, involve all input tokens in this calculation. We noticed that this makes the GT loss more significant in the decoder model in many cases. We will revise our description to make this point clearer.
>
>
> **Comments 5) Comments on writing & limitation (Minor comments 1-4, Limitation)**
>
> **Answer 5)** Thank you for examining the complexities of our paper. We will address the points you've raised, including issues related to the LLaMA-7B model, equation notation, causal attention, and writing style in section 3.3. Additionally, we will consider the weaknesses you've highlighted and update the limitation section in the revised version.

---

> > ### Comment · Reviewer_cozw · 2023-08-13
> >
> > I want to thank the authors for their response. Unfortunately I can't see the tables with the new results. In addition, while perplexity is fine, I'd love to see the simple step of actual results on the downstream tasks.

---

> > > ### Author Response · Authors · 2023-08-13
> > >
> > > Thank you very much for your valuable feedback. Respectfully, starting from this rebuttal, we are allowed to attach a **1-page PDF** in a **global response** section. You can kindly refer to the table in the attached PDF. I regret that I didn't mention this in the rebuttal response section. To address this, I will include the results for the NLU and CSQA downstream task accuracy results in this comment. (more detailed results can be found in the attached PDF of the global response)
> > >
> > > **Table R1**: Accuracy results for the OPT-1.3B model fine-tuned on natural language understanding GLUE tasks using different QAT-KD methods on W4A16 and W2A16. ACC metrics are averaged based on five random seed tests.
> > >
> > > | Precision | QAT KD Method | CoLA ACC (↑) | MRPC ACC (↑) | SST-2 ACC (↑) | RTE ACC (↑) |
> > > |-----------|---------------|--------------|--------------|---------------|-------------|
> > > | OPT-1.3B FP16 | - | 61.03 | 81.92 | 94.26 | 76.53 |
> > > | W4A16 PTQ | OPTQ-CW | 54.61 | 80.14 | 95.07 | 56.32 |
> > > | - | AWQ-CW | 13.63 | 66.42 | 94.26 | 54.51 |
> > > | W4A16 QAT | Logit [20] | 50.76 | 81.94 | 93.57 | 75.23 |
> > > | - | GT+Logit | 54.07 | 83.17 | 93.34 | 75.31 |
> > > | - | **TSLD** | **56.33** | **83.33** | **94.05** | **75.97** |
> > > | W2A16 QAT | Logit [20] | 48.72 | 81.62 | 93.08 | 74.15 |
> > > | - | GT+Logit | 50.10 | 82.10 | 92.77 | 73.79 |
> > > | - | **TSLD** | **54.47** | **82.20** | **93.92** | **75.31** |
> > >
> > > **Table R3**: Accuracy (ACC_norm) results of the OPT-2.7B model fine-tuned on the CSQA (Commonsense QA) with different W2A16 QAT-KD methods.
> > > | QAT KD (W2A16)   | PIQA (↑)  | OpenbookQA (↑) | ARC_easy  (↑) | ARC_challenge  (↑) | GSM8K (↑) |
> > > |:----------------|:--------------|:-------------------|:-----------------|:-----------------------|:--------------|
> > > | OPT-2.7B FP16   | 76.71         | 49.60              | 66.12            | 37.20                  | 20.39         |
> > > | Logit [20]     | 74.32         | 45.40              | 58.92            | 31.91                  | 20.02         |
> > > | GT+Logit        | 74.97         | 46.20              | 58.84            | 32.16                  | 19.56         |
> > > | **TSLD**            | **75.62**     | **46.81**          | **59.39**        | **33.45**              | **20.24**     |
> > >
> > > **Table R4**: Perplexity and accuracy results of W2A16 QAT-KD on PTB task and text generation-based reasoning tasks for various GLM models (GPT-Neo-1.3B, LLaMA-7B)
> > >
> > > | Model (W2A16)   | GPT-Neo-1.3B  PPL (↓) | OPT-6.7B GSM8K  (↑) | LLaMA-7B PTB (↓) | LLaMA-7B GSM8K  (↑)  |
> > > |:---------------|:---------------------|:-------------------|:-----------------|:-------------------|
> > > | FP16           | 17.62                | 22.52              | 9.10              | 32.02              |
> > > | Logit [20]          | 21.01                | 21.08              | 10.82            | 25.47              |
> > > | **TSLD**           | **19.27**            | **24.49**          | **10.79**        | **26.23**          |
> > >
> > > In the three tables presented, it is evident that the TSLD method not only improves perplexity but also significantly enhances downstream accuracy. This can be observed across NLU (Table R1), and reasoning (CSQA) tasks(Table R3/4).
> > >
> > > If there are further points to discuss or valuable feedback to share, we would always be grateful and welcoming. We would greatly appreciate it if you could consider the various downstream task accuracy results we have prepared during the rebuttal.

---

### Official Review · Reviewer_QKmL · 2023-06-30

**Soundness:** 3 good
**Presentation:** 2 fair
**Contribution:** 2 fair
**Rating:** 6
**Confidence:** 4

**Summary:**

This paper proposes to improve the quantization of generative language models via an adaptively scaled logit distillation method.

With an intuitive illustration in Figure 1 and validation results with attention heatmap discrepancy, the paper identified a cumulative quantization error introduced by the masked attention, i.e., later tokens attend to more tokens before, leading to a larger quantization error.

Furthermore, the paper investigated the characteristics of Layer-to-Layer (L2L) distillation (aligning output activations and attention heatmaps) and the logit distillation from teacher models. Results show these two methods differ from the inner representations and min-max dynamic ranges. This result motivates a proposal of Token-Scaled Logit Distillation, where tokens with higher confidence of teacher predictions are down-weighted in the distillation to avoid overfitting.

Experimental results on GPT-2 and OPT-series show that the proposal effectively maintains the generation quality (lower perplexity compared with baseline methods) and reasoning ability of the original full precision model (better accuracy on GSM7k).



**Strengths:**


-  Exploring the quantization techniques for billion-level language models is a vital problem.

- The finding of cumulative quantization errors of mask attention in Transformer models is interesting.


**Weaknesses:**

The paper can benefit from addressing the following weaknesses:

- Paper Organization: The current paper lacks clarity in its organization, making it challenging to follow the logical flow between sections. Improving the coherence and connection between sections would greatly enhance the reading experience.

- Integration of Challenges: The paper identifies two challenges in quantizing LLMs, namely the cumulative quantization errors of masked attention and the necessity of ground-truth loss. However, it would be more effective to investigate these challenges independently. Additionally, the proposed method seems less relevant to the first finding, which, in my opinion, is the most crucial problem in the quantization process.

- Lack of Clear Motivation: The motivation behind the proposed dynamic weighting method, aimed at addressing the potential overfitting problem, is not adequately explained. The evidence presented in Figure 3(b) is insufficient to support the claim that "(high confidence) logit distillation might mirror the effects of ground truth loss, thereby contributing to the observed overfitting." It would be beneficial to explore whether using ground truth alone leads to overfitting, as well as whether removing distillation of high-confidence tokens can mitigate the overfitting problem. Additionally, justifying the up-weighting of low-confidence tokens, which exhibit high cross-entropy loss and could potentially represent incorrect predictions from the teacher models, is crucial to convince readers of the proposal's validity.

**Questions:**

- In Figure 2(b) Left, why is the bottom left corner (where tokens attend fewer tokens ?) of the heatmap also bright?
- In Figure 2(b) Right, what is the reason for deeper layers exhibiting lower attention MSE error in the decoder?
- Original QuantGPT incorporates Token-level Contrastive Distillation and Module-dependent Dynamic Scaling, did you solely utilize logit distillation?

---

> ### Author Rebuttal · Authors · 2023-08-10
>
> We are grateful for the thorough and thoughtful reviews provided by the reviewer. Below are our responses to the raised questions.
>
> **Comment 1) Paper organization (Weakness 1)**
>
> **Answer 1)** We concur with the reviewer's feedback regarding the shortcomings in the logical flow, structure, and coherence of our main contents. Acknowledging reviewer's remarks, we will enhance the paper's organization, in particular by revising Sec.3.3 and Sec. 4, to clarify our key motivations and claims.
>
>
> **Comment 2) Integration of challenges: Connection between cumulative quantization errors and Logit KD (Weakness 2)**
>
> **Answer 2)** We thank the reviewer for highlighting ambiguities between our observation of cumulative quantization errors and our Logit-KD proposal. In summary, we've unveiled 1) distinct error characteristics in Transformer decoder quantization not addressed by previous L2L KD, and 2) Logit KD's unique error compensation mechanism, which imposes flexibility in adjusting magnitude across middle layers to align the final logit, thus enabling accurate text generation.
>
> We've prepared Fig. R1 (attached pdf) to clarify our findings:
> - Fig. R1(a): Transformer decoders show cumulative quantization errors at attention maps from token zero to N-1, resulting in notable accuracy loss. This increasing error magnitude is universal, and Fig. R1(a) aligns with Fig.2(b) from our paper.
>
> - Fig. R1(b) and (c): Applying L2L KD (state-of-the-art Transformer encoder quantization [33],[13],[14]) failed to overcome these errors, causing significant performance degradation (Table 1) and memory inefficiency (Table R6). L2L KD reduced attention map errors (Fig. R1(c)) but not the final logit distance (Fig. R1(b)), which most affects text generation accuracy. Despite prior success, L2L KD proved ineffective for Transformer decoders.
>
> - Fig. R1(b) and (d): Recognizing the final logit's importance, we used Logit KD to target finetuning solely on it (without layer-wise regularization). Although attention map error reduction was modest (Fig. R1(d)), logit distance was sharply reduced thanks to Logit KD (Fig. R1(b)), enhancing Transformer decoder performance (Table 1).
>
> - Fig. R1(e): We discovered Logit KD's success lies in allowing intermediate tokens to vary their attention values across channels (Fig. R1(e)-Left), diverging from full-precision (increasing attention map error as in Fig. R1(d)) in the middle layers while aligning the last layer output with the full-precision model (Fig. R1(e)-Right). This focus on the final logit instead of layer-wise regularization leads to more robust text generation.
>
> **Comment 3) Integration of clear motivation: Empirical evidence of overfitting issues with ground truth (GT) loss effectiveness of TSLD (Weakness 3)**
>
> **Answer 3)** We appreciate the reviewer's request for empirical evidence concerning overfitting issues with GT loss and the efficacy of TSLD in addressing them. We've added Fig.R1(f) to furnish this evidence. Specifically, in response to the reviewer's comment, we broke down TSLD into two components, TSLD_down and TSLD_up, and conducted an ablation study to analyze the method.
> - GT loss causes significant overfitting and Logit KD alone cannot rectify. TSLD_down addresses overfitting from GT loss by reducing the influence of high-confidence token predictions. By implementing this scale in logit distillation, we significantly reduced the overfitting that was previously detected in both the GT-only and GT+Logit cases.
> - TSLD_up places additional emphasis on distillation for low-confidence token predictions. By adding TSLD_up to GT+TSLD_down, we observed a noticeable decrease in evaluation loss. This underlines the vital role of distillation in enriching low-confidence predictions with soft label information, thus improving QAT performance.
> - The benefits of TSLD_down and _up stem from their ability to reduce logit distance more effectively than Logit KD, a key factor in successful text generation (Fig.R1(b)).
>
>
> **Comment 4) Questions regarding Fig. 2(b) (Question 1, 2)**
>
> **Answer 4)** We've plotted an additional attention map loss in Fig.R2(a) to address the reviewer's inquiry, with the following details:
> - We measured the masked attention error for each token within the attention map by normalizing the error for each token row by the number of tokens attended to. This normalized error is represented as a heatmap on the left, with averaged errors per token plotted on the right by token index.
> - Though the trend of increasing error magnitude is consistent, variations can be seen in the heatmap patterns between Fig.2(b) and Fig. R1(a). Likewise, while the cumulative error trend is universal, the error magnitude patterns across layers may differ; the 8th layer shows the lowest magnitude in Fig.2(b), while the 0th layer shows the smallest magnitude in Fig. R1(a).
>
>
> **Comment 5) Difference in quantization settings between our Logit KD and QuantGPT [20] (Question 3)**
>
> We appreciate the opportunity to clarify the difference in quantization settings between our Logit KD and QuantGPT[20], specifically regarding token-level contrastive distillation and module-dependent dynamic scaling (PACT+).
>
> QuantGPT's token-level contrastive distillation uses random negative sample tokens throughout the sequence, a selection that can potentially cause performance degradation and issues with robustness. This issue has been acknowledged in the revised QuantGPT paper (https://arxiv.org/pdf/2203.10705.pdf). To ensure a fair comparison, we omitted the contrastive loss from our implementation of Logit KD.
>
> We also experimentally found that PACT+ does not provide meaningful gain compared to our TWN-based ternary quantizer. Table R5 shows a comparison between PACT+ and ours; no significant performance differences were noted across PTB and Wikitext-2 tasks, with the TWN method actually performing slightly better. Thus, we chose the TWN-based ternary quantizer for our comparative experiments.

---

> > ### Comment · Reviewer_QKmL · 2023-08-16
> >
> > Thanks for your detailed response and experimental evidence.
> >
> > - Regarding Answer 1 and Answer 2: The connection between cumulative quantization errors and Logit KD looks much more evident to me.
> >
> >
> > - Regarding Answer 3: The decomposition Fig. R1(f) is very helpful, making the overfitting of groud-truth training clearer and the effect of dynamically reweighting.
> >
> > Overall, I think the response addressed my concerns well. I am happy to raise my score to 6, and I hope these discussions and results can be integrated into the revision as well.

---

> > > ### Author Response · Authors · 2023-08-16
> > >
> > > We are grateful to the reviewer for reviewing our responses with materials and providing us with feedback. Your constructive and insightful comments have been invaluable in guiding our efforts to improve the manuscript. We will make sure to incorporate your valuable suggestions into the revised manuscript.
> > >
> > > Once again, we deeply appreciate your invaluable review.

---

### Official Review · Reviewer_aiZD · 2023-07-06

**Soundness:** 3 good
**Presentation:** 3 good
**Contribution:** 3 good
**Rating:** 6
**Confidence:** 4

**Summary:**

This paper proposed the token-scaled logit distillation method to counteract accuracy loss caused by using Quantization-Aware Training（QAT） methods for generative language models . (1) The author proposed the logit distillation method to overcome the limitation in token prediction recovery by modifying the intermediate representation. (2) the author proposed an adaptive KD method that adjusts Logit KD based on token confidence to prevent the overfitting problem caused by the combination of Logit KD and ground truth loss in QAT.

**Strengths:**

1、The idea of adjusting Logit KD based on token confidence to prevent the overfitting problem is novel.

2、The method used in this paper is generalized and applicable to models of different sizes and different types of text generation tasks, including language modeling and reasoning.

3、The experiments involves multiple LLMs with various model scale (0.1B,2.7B,6.7B), which makes the main experiments solid and convincing.

**Weaknesses:**

1、The FP 32 baseline used in the comparison experiment is not explained. And the experiment only uses one baseline, the proof is not sufficient.

2、 In addition, the experiment selected one dataset for each tasks for validation, and the number of datasets is small. We recommended to add SVAMP and CSQA datasets for inference tasks, and SGD datasets for text generation tasks.

3、The experiments include only main results (comparing with baselines) and the ablation studies. It would be better to conduct more detailed analyses or qualitative analysis to verify the proposed methods (e.g. whether the issues proposed in the motivation are fixed).

**Questions:**

None

---

> ### Author Rebuttal · Authors · 2023-08-10
>
> We sincerely appreciate the insightful feedback provided by the reviewers. In response to the raised questions, our answers are as follows.
>
> **Comment 1) Clarification of Baseline Settings and Expanding TSLD Evaluation Across Tasks and Models (Weakness 1, 2)**
>
> **Answer 1)**
>
> FP 32 baseline settings: ​​​​ Our primary focus lies in recovering the performance of a fine-tuned model using QAT after fine-tuning a pre-trained model for a specific task. Therefore, we first fine-tune the open-sourced pre-trained model for the target task to establish an FP baseline.
>
>
> Expanding TSLD Evaluation: To address the concern about the limited datasets we initially used, we've expanded our evaluation to include 4 additional NLU tasks and 4 CSQA tasks. Furthermore, to expand the number of baselines of our experiments, we extended our comparative experiments to other GLM models, GPT-Neo-1.3B and LLaMA-7B.
>
> The experimental results are detailed in Tables R1-R5 of the attached PDF. The lessons derived from these experimental results are as follows:
>
> - **W4A16 OPT-1.3B on NLU and CSQA (Tables R1,R3)**: 4-bit TSLD consistently exhibits significantly more robust performance compared to OPTQ and/or AWQ, the latest SOTA PTQ [9][R1] techniques (we used the code published by the authors of corresponding papers). In particular, TSLD achieved the lowest perplexity and the highest accuracy across all the experiments except SST-2, where its accuracy is in-par with the full-precision case.
>
> - **W2A16 OPT-1.3B on NLU and CSQA (Tables R1,R2,R3)**: ternary TSLD consistently outperforms the alternative QAT methods for all the cases tested, demonstrating its superior performance in bridging the accuracy gap with the full-precision cases. Interestingly, ternary TSLD even achieved similar or superior accuracy compared to 4-bit PTQ in many tasks, highlighting its benefits on both accuracy and memory savings (W4A16-761.9MB -> W2A16-473.2MB).
>
> - **TSLD on GPT-Neo-1.3B and LLaMA-7B (Table R4)**: We tested the general applicability of TLSD on popular open-sourced GLM models (GPT-Neo-1.3B, LLaMA-7B) using the PTB and GSM8K datasets. Our findings indicate that TSLD consistently surpassed the competitor, Logit KD [20, QuantGPT]. Particularly in the GSM8K dataset for LLaMA-7B, TSLD equaled the perplexity while attaining higher accuracy, demonstrating its capability to mitigate overfitting issues. However, we observed that ternary weight quantization posed greater challenges on LLaMA-7B, a similar phenomenon also noted in [R1]. This difficulty seems to correlate with the broader weight range found in LLaMA-7B. As a result, we are exploring methods to tune the diverse weight distributions of GLMs amenable for quantization, as an interesting future research.
>
> **Comment 2) Detailed and Qualitative Analysis of TSLD Method (Weakness 3)**
>
> **Answer 2)** We aim to provide a detailed analysis connecting each challenge with our analysis and the proposed method.
>
> Starting with cumulative quantization error, we've unveiled 1) distinct error characteristics in Transformer decoder quantization not addressed by previous L2L KD, and 2) Logit KD's unique error compensation mechanism, which imposes flexibility in adjusting magnitude across middle layers to align the final logit, thus enabling accurate text generation.
>
> We've prepared Fig. R1 (attached pdf) to clarify our findings:
> - Fig. R1(a): Transformer decoders show cumulative quantization errors at attention maps from token zero to N-1, resulting in notable accuracy loss. This increasing error magnitude is universal, and Fig. R1(a) aligns with Fig.2(b) from our paper.
>
> - Fig. R1(b) and (c): Applying L2L KD (SOTA Transformer encoder quantization [33],[13],[14]) failed to overcome these errors, causing significant performance degradation (Table 1) and memory inefficiency (Table R7). L2L KD reduced attention map errors (Fig. R1(c)) but not the final logit distance (Fig. R1(b)), which most affects text generation accuracy. Despite prior success, L2L KD proved ineffective for Transformer decoders.
>
> - Fig. R1(b) and (d): Recognizing the final logit's importance, we used Logit KD to target finetuning solely on it (without layer-wise regularization). Although attention map error reduction was modest (Fig. R1(d)), logit distance was sharply reduced thanks to Logit KD (Fig. R1(b)), enhancing Transformer decoder performance (Table 1).
>
> - Fig. R1(e): We discovered Logit KD's success lies in allowing intermediate tokens to vary their attention values across channels (Fig. R1(e)-Left), diverging from full-precision (increasing attention map error as in Fig. R1(d)) in the middle layers while aligning the last layer output with the full-precision model (Fig. R1(e)-Right). This focus on the final logit instead of layer-wise regularization leads to more robust text generation.
>
> Concerning overfitting issues with GT loss and the efficacy of TSLD in addressing them, We've added Fig.R1(f) to furnish this evidence. Specifically, we broke down TSLD into two components, TSLD_down and TSLD_up, and conducted an ablation study to analyze the method.
> - GT loss causes significant overfitting and Logit KD alone cannot rectify. TSLD_down addresses overfitting from GT loss by reducing the influence of high-confidence token predictions. By implementing this scale in logit distillation, we significantly reduced the overfitting that was previously detected in both the GT-only and GT+Logit cases.
> - TSLD_up places additional emphasis on distillation for low-confidence token predictions. By adding TSLD_up to GT+TSLD_down, we observed a noticeable decrease in evaluation loss. This underlines the vital role of distillation in enriching low-confidence predictions with soft-label information, thus improving QAT performance.
> - The benefits of TSLD_down and _up stem from their ability to reduce logit distance more effectively than Logit KD, a key factor in successful text generation (Fig.R1(b)).

---

### Official Review · Reviewer_kT5b · 2023-07-06

**Soundness:** 3 good
**Presentation:** 4 excellent
**Contribution:** 3 good
**Rating:** 7
**Confidence:** 5

**Summary:**

This research paper focuses on Generative Language Models (GLMs) and the challenges of their deployment due to their large model sizes. The authors introduce a novel approach to tackle the challenges faced by Quantization-Aware Training (QAT) in fine-tuned GLMs.

**Problem**: GLMs, despite their impressive performance in text generation, understanding, and reasoning, face deployment challenges due to their large model sizes. Quantization techniques like Post-Training Quantization (PTQ) and Quantization-Aware Training (QAT) have been used for model compression but result in significant decreases in accuracy.

**Approach**: The authors propose a new approach, Token-Scaled Logit Distillation (TSLD), to enhance QAT for sub-4bit QAT inference. TSLD is a knowledge distillation technique specifically designed for GLMs. It counters the issues of quantization errors and overfitting, prevalent in existing QAT methods. The method is based on token-scaled logit distillation, which prevents overfitting and allows better learning from the teacher model and ground truth.

**Results**: When TSLD was applied to a variety of GLMs, including a 6.7 billion model for the first time, it achieved comparable, if not superior, performance in text generation on ternary and 4-bit inference. On an arithmetic reasoning task, TSLD even achieved task accuracy at least on par, if not better, with other methods.

Experiments were performed on two tasks: language modeling (using the PTB dataset) and a reasoning-based text-generation task (using the GSM8K dataset). The models were trained using the HuggingFace fine-tuning repository, with varying learning rates and sequence lengths depending on the model's size. They also implemented a QAT-based KD framework using PyTorch's Pipe API for pipeline parallelism.

Results showed that TSLD achieved the lowest Perplexity (PPL), signifying it performed the best in both language modeling and reasoning tasks, outperforming other methods such as Post-Training Quantization (PTQ), QAT KD, and others. TSLD's performance allowed QAT models to match full-precision performance across different model capacities.

Regarding sub-4 weight quantization, the researchers found that L2L KD couldn't run on larger GLMs due to memory constraints. TSLD, however, outperformed other QAT approaches across all model sizes, maintaining PPL degradation of no more than 1.0 from the baseline.

For the reasoning task, TSLD was shown to effectively balance language modeling and reasoning performance, achieving excellent reasoning accuracy while maintaining competitive perplexity. Ablation study also suggested that sub-4bit quantization with TSLD considerably speeds up the inference of large models while maintaining competitive model quality.

**Conclusion**: The successful results underline the potential of the proposed TSLD method in facilitating the deployment of ultra-low precision GLMs. The method allows for the practical deployment of large-scale GLMs with less than 1.0 degradation in perplexity and no loss of accuracy in a reasoning task.

**Strengths:**

**Innovative Method**: The paper introduces the Token-wise Scaling for Low-bit Distillation (TSLD) method for Quantization-Aware Training (QAT), a novel approach that seems to substantially improve the performance of low-precision models. And this a very early work about Ternary quantization and sub-4 quantization of GLMs.

**Detailed Implementation information**: The paper provides clear and comprehensive details on the experimental setup, including task and model selection, training settings, and implementation settings. This thoroughness makes the results more credible and the experiments reproducible by other researchers.

**Clean and accurate formulation**: the formulation process is clean and accurate, to the best of my knowledge.

**Practical Results**: The paper demonstrates practical advantages of TSLD, such as matching full-precision model performance across various model sizes and maintaining low perplexity degradation with sub-4 weight quantization. These findings suggest that the TSLD method is not only theoretically promising but also practically beneficial, paving the way for low-precision inference in real-world applications.

**Weaknesses:**

Lack of Broader Evaluation: The paper focuses primarily on GLMs like GPT-2 and OPT for the experiments. This limited scope might not fully illustrate how the TSLD method performs with other types of GLM models. Specifically, if the evaluation tasks can be more various and abundant, it would be very helpful

Specificity of Tasks: While the paper does test the proposed method on different tasks, it mainly focuses on the PTB language modeling task and GSM8K reasoning task. An evaluation of the method's effectiveness on a broader array of tasks, or in specific real-world applications, could have added to the robustness of the study. (My concern comes from: perplexity is very insensitive, and arithmetic reasoning dataset GSM8K is not large. So "less than 1.0 perplexity degradation and preserving arithmetic reasoning task accuracy" is not that convincing.  it is really recommended to have more evaluation on other tasks)

**Questions:**

1. About the TSLD methods details "Token-Scaled Logit Distillation (TSLD). Based on investigations into the probabilistic relation of token predictions and overfitting in QAT, we propose an adaptive KD method that adjusts Logit KD based on token confidence. This approach utilizes the phenomenon of confidence disparity in token predictions from the teacher model. Our method, called TSLD, de-emphasizes Logit KD for high-confidence tokens to prevent overfitting in QAT while emphasizing logit distillation for low-confidence tokens with high entropy probability distributions."
I am eager to know, how much additional computing are required in the TSLD process, such as incorporating the token confidence into the distillation process?

2. Without specific designed hardware, how much advantage could TSLD powered Ternary Weight quantization have, over the 4-bit quantization of LLMs? With what kind of designed hardware, how much advantage could TSLD powered Ternary Weight quantization have, over the 4-bit quantization of LLMs? The clarity regarding this point will illustrate the practical significance of this work.

**Limitations:**

**Hardware Restrictions**: The TSLD method uses simple ternary weight quantization, which is designed to be hardware-friendly. However, this means that the effectiveness of TSLD may be largely dependent on the specific hardware it's being implemented on. Some hardware configurations might not support ternary or lower precision computations efficiently, which could limit the usefulness or performance of TSLD in such environments.

---

> ### Author Rebuttal · Authors · 2023-08-10
>
> We sincerely appreciate the insightful feedback provided by the reviewer. In response to the raised questions, our answers are as follows.
>
> **Comments 1) Expanding TSLD Evaluation (Weakness)**: To address the concern about the limited datasets we initially used, we've expanded our evaluation to include 4 additional NLU tasks and 4 CSQA tasks. Furthermore, to expand the number of baselines of our experiments, we extended our comparative experiments to other GLM models, GPT-Neo-1.3B and LLaMA-7B.
>
> The experimental results are detailed in Tables R1-R5 of the attached PDF. The lessons derived from these experimental results are as follows:
>
> - **W4A16 OPT-1.3B on NLU and CSQA (Tables R1,R3)**: 4-bit TSLD consistently exhibits significantly more robust performance compared to OPTQ and/or AWQ, the latest SOTA PTQ [9][R1] techniques (we used the code published by the authors of corresponding papers). In particular, TSLD achieved the lowest perplexity and the highest accuracy across all the experiments except SST-2, where its accuracy is in-par with the full-precision case.
>
> - **W2A16 OPT-1.3B on NLU and CSQA (Tables R1,R2,R3)**: ternary TSLD consistently outperforms the alternative QAT methods for all the cases tested, demonstrating its superior performance in bridging the accuracy gap with the full-precision cases. Interestingly, ternary TSLD even achieved similar or superior accuracy compared to 4-bit PTQ in many tasks, highlighting its benefits on both accuracy and memory savings (W4A16-761.9MB -> W2A16-473.2MB).
>
> - **TSLD on GPT-Neo-1.3B and LLaMA-7B (Table R4)**: We tested the general applicability of TLSD on popular open-sourced GLM models (GPT-Neo-1.3B, LLaMA-7B) using the PTB and GSM8K datasets. Our findings indicate that TSLD consistently surpassed the competitor, Logit KD [20, QuantGPT]. Particularly in the GSM8K dataset for LLaMA-7B, TSLD equaled the perplexity while attaining higher accuracy, demonstrating its capability to mitigate overfitting issues. However, we observed that ternary weight quantization posed greater challenges on LLaMA-7B, a similar phenomenon also noted in [R1]. This difficulty seems to correlate with the broader weight range found in LLaMA-7B. As a result, we are exploring methods to tune the diverse weight distributions of GLMs amenable for quantization, as an interesting future research.
>
>
> **Comments 2) Analysis of Computation Requirements of TSLD (Question 1)**
>
> The TSLD method, which incorporates token cross-entropy loss into logit distillation, requires two distinctive operations.
>
> $L_{TSLD} = L_{GT} + \sum_{n=1}^{N} \left( \text{softmax} \left( \sum_{i=1}^{V} y_{n,i} \log(P_{n,i}^{T}) \right) \circ \sum_{i=1}^{V} P_{n,i}^{T} \log(P_{n,i}^{S}) \right)$
>
> Specifically, given that \( N \) represents the entire sequence length and \( V \) is the vocabulary size, the term $ \sum_{i=1}^{V} y_{n,i} \log(P_{n,i}^{T}) $ computes the cross-entropy loss using the teacher logits. This result is then passed through a softmax function to derive the scaling value for each token. This scale value is then element-wise multiplied with   $\sum_{i=1}^{V} P_{n,i}^{T} \log(P_{n,i}^{S})$, which is token-wise logit distillation. (note that typical logit distillation would multiply the token-specific logit distillation value by ${1/N}$).
>
> Since TSLD leverages the final logit values of the teacher model used in logit distillation for dynamic scale values, no additional memory consumption is required. The computations for the teacher model's cross-entropy loss and the softmax computation for scale values have a complexity of ${O(N)}$, making the overhead of TSLD negligible in terms of training latency.
>
> To assess the efficiency of TSLD, we report the training speeds, and GPU memory consumption, for various QAT-KD methods, as shown in Table R7. When comparing the TSLD method to other Logit distillation-based methods, we found that there's no additional memory consumption, and the training speed either slightly decreases or remains unaffected. Such outcomes underscore the advantages of the memory-efficient TSLD; it elevates QAT-KD performance while consuming memory similar to logit distillation, which uses significantly less memory compared to the memory-intensive L2L distillation.
>
> **Comment 3) Specific Hardware for Ternary Weight Quantization (Question 2, Limitations)**
>
> **Answer 3)** Ternary quantization, representing weights as {+1, 0, -1}, is an appealing method for accelerating matrix multiplication by transforming multiplication into simpler additions. This approach enables a "multiplication-less" implementation, reducing the computational complexity of matrix multiplication, a universal operation for deep learning models. The weights determine whether corresponding activations are added (w=1), subtracted (w=-1), or ignored (w=0), bypassing the need for complex hardware logic for multiplication. Examples of ternary weight quantization applications include:
> - Custom Accelerator: [R2] utilized ternary weight quantization in FPGA to streamline computational logic for custom acceleration, showcasing superior performance compared to equivalent GPU implementations.
> - CPU Implementation: [R3] showed how ternary weight quantization can be applied in CPUs, not just custom accelerators, achieving a 2 to 4x speedup in inference running on off-the-shelf CPUs using sub-word SIMD instructions.
> - In-memory Computing: [R4] was a pioneer in incorporating ternary weight quantization into in-memory computing. By representing weights as {-1, 0, 1}, this method significantly simplified the in-memory processing implementation, enhancing the performance of DNN inference.
>
> [R2] Nurvitadhi et al, “Can FPGAs Beat GPUs in Accelerating Next-Generation Deep Neural Networks?” FPGA 2017
> ​​[R3] Choi et al, “​​TernGEMM: GEneral Matrix Multiply Library with Ternary Weights for Fast DNN Inference” SIPS 2021
> [R4] Jain et al, “TiM-DNN: Ternary in-Memory accelerator for Deep Neural Networks” VLSI 2020

---

> > ### Comment · Reviewer_kT5b · 2023-08-16
> > **Thanks for addressing my questions**
> >
> > Thanks for addressing my questions. I keep the original rating as 7.
> >
> > If the code or even only model could be provided through an anonymous link, the reproducibility of this work will improve.

---

> > > ### Author Response · Authors · 2023-08-16
> > >
> > > Thank you for your feedback. We will ensure that your insightful and valuable comments are incorporated into our final manuscript.
> > >
> > > Furthermore, based on your comments, we have prepared an anonymized code repository. The refined version will be officially released alongside our final manuscript.
> > >
> > > https://anonymous.4open.science/r/tsld_anonymized-D6D1
> > >
> > > Once again, we deeply appreciate the time and effort you've dedicated to feedback on our manuscript.

---

> > > > ### Comment · Reviewer_kT5b · 2023-08-17
> > > > **Thanks very much for the reply**
> > > >
> > > > Thanks very much for the timely reply!
> > > >
> > > > The anonymized code link makes me more confident of the current rating at least 7. Could you share some model weights also in anonymized link if allowed? After examine the result, I might increase my rating further.
> > > >
> > > > Best regards

---

> > > > > ### Author Response · Authors · 2023-08-18
> > > > > **Thanks very much for your further interest**
> > > > >
> > > > > We greatly appreciate your time in reviewing the code we provided and thank you for your further interest in our approach.
> > > > > To share the QAT model weight where our method is applied, we have prepared an anonymous link as follows:
> > > > >
> > > > > link: https://we.tl/t-NH1wysJtqW
> > > > > (Accept cookies/terms of service required before downloading, expires in 7 days)
> > > > >
> > > > > **About Model Weight**
> > > > >
> > > > > Due to the capacity limits of the anonymous file link, we are sharing the weight files of the two models where **two QAT-KD methodologies (Logit+GT, TSLD)** have been applied to the **OPT-125m PTB fine-tuned model** (Table1). In the future, we plan to share model weights (through huggingface) for verification on various billion-scale fine-tuned models.
> > > > >
> > > > > **Guideline for Model Weight Examination**
> > > > >
> > > > > For the convenience of the reviewer's examination of the model weight, here's how the model performance verification proceeds:
> > > > >
> > > > > - We evaluated the QAT model's PPL performance using the latest version of huggingface (Transformers) language-modeling example code (https://github.com/huggingface/transformers/tree/main/examples/pytorch/language-modeling). We have re-confirmed that the evaluation results in the official huggingface environment match the outcomes from our experimental setup. (Logit+GT PPL: 21.91, **TSLD PPL: 18.97**)
> > > > >    - Note that the model file we shared corresponds to the 40 Epoch QAT results conducted in the ablation study of Figure R1 (f) in the rebuttal. By increasing the number of epochs, you can obtain the same results presented in Table 1 of the manuscript.
> > > > >
> > > > > - The shared model folder's file configuration is compatible with the **huggingface AutoModel** method. Specifically, by simply placing the names of the two model folders (`opt_ptb_40_2bit_logit_gt`, `opt_ptb_40_2bit_tsld`) into the `--model_name_or_path` argument of the main code, you can easily build ternary weight OPT-125m model.
> > > > >
> > > > > **Details**
> > > > >
> > > > > - The direction of the QAT-KD in this paper involves applying QAT to *task-specific fine-tuned GLM*. Therefore, please note that if the evaluation is conducted on a task that wasn't fine-tuned, the performance might not be satisfactory.
> > > > > - The model weights have per-tensor ternary quantized values stored (ex. [0, 1.7, -1.7, 0, 1.7, 0, -1.7 ... ]), but they aren't actually encoded in the INT2 type. Please note this. We plan to implement the INT2 datatype in the future for the sake of saving actual model size.
> > > > > - In each model folder, there are files named `training_log.json` and `eval_log.json`. These contain logs of training loss, evaluation loss, and epoch-by-epoch evaluation PPL results during the QAT-KD training. You can refer to these to examine how QAT training has proceeded.
> > > > > - `all_results.json` contains the model's last evaluation PPL performance and its best PPL performance. By comparing the two results from the `opt_ptb_40_2bit_logit_gt` model, you can re-confirm instances of overfitting, as seen in the logit + GT QAT.
> > > > >
> > > > > We would like to express our gratitude once again for the interest the reviewer has shown in our method. We hope that the shared model weights and guides will assist in further examination. Should there be any additional queries or difficulties during the verification process, please do not hesitate to leave a comment. We welcome any feedback.

---

### Official Review · Reviewer_EGP4 · 2023-07-12

**Soundness:** 2 fair
**Presentation:** 3 good
**Contribution:** 2 fair
**Rating:** 5
**Confidence:** 4

**Summary:**

The article introduced a new approach for LLM Quantization Aware Training (QAT).

**Strengths:**

The paper is well written and the proposed idea is described clearly.


**Weaknesses:**

I have some concerns about the evaluation in the paper. I

First, the number of tasks used for accuracy/perplexity comparison between float and the proposed method is too small. The authors only evaluate the proposed method on two small datasets. I would encourage the authors to include more datasets in the evaluation, such as those Q&A tasks and SuperGlue tests datasets listed in the GPT-3 paper. This would help to provide more evidence to support the claims made in the paper: the proposed method counteract the issues of accuracy loss.

Second, the authors only report accuracy and perplexity results for the int2 data type. It would be great for the readers to see the accuracy and perplexity numbers for all data types (int8, int4, and int2).  The authors do provide latency benchmarks for the int8 data type, but they do not report any quality results. I would encourage the authors to report accuracy and perplexity results for all data types so that the readers can have a better understanding of the trade-offs between accuracy, latency, and memory usage.

Here are some specific suggestions for how the authors could improve the evaluation:

1. Include more tasks in the evaluation, such as those Q&A tasks and SuperGlue tests datasets listed in the GPT-3 paper.
2. Report accuracy and perplexity results for all data types (int8, int4, and int2).
3. Conduct evaluation to assess the quality of instruction tuning dataets.




**Questions:**

1. It's not clear to me whether table 1 shows the training or the test perplexity for the PTB dataset.

2. Why only results from W2A32 are shown in table 2? Can you please also include W4A32 ones?


**Limitations:**

Need more evaluation results to justify the contributions.

---

> ### Author Rebuttal · Authors · 2023-08-10
>
> We appreciate the reviewer’s valuable feedback. Responses to your comments are provided below.
>
> **Comment 1) Expanding TSLD Evaluation Across NLU/CSQA Tasks and Models**
> >Weakness 1) “the number of tasks used for accuracy/perplexity comparison between float and the proposed method is too small. The authors only evaluate the proposed method on two small datasets. I would encourage the authors to include more datasets in the evaluation.”
> Weakness 3) “Include more tasks in the evaluation”
> Question 2) “Can you please also include W4A32 ones?”
> Limitation) “Need more evaluation results to justify the contributions.”
>
> **Answer 1)**
>
> Thank you for the constructive feedback on our paper's evaluation approach. To address the concern about the limited datasets we initially used, we've expanded our evaluation to include 4 additional NLU tasks and 4 CSQA tasks. Furthermore, to ensure the robustness and generalizability of our method, we extended our comparative experiments to other GLM models, GPT-Neo-1.3B and LLaMA-7B. (We added experimental results as best we could, time permitting.)
>
> [R1] Lin et al. “AWQ: Activation-aware Weight Quantization for LLM Compression and Acceleration”, Arxiv 2023.
>
> The experimental results are detailed in Tables R1-R5 of the attached PDF. The lessons derived from these experimental results are as follows:
>
> - W4A16 OPT-1.3B on NLU and CSQA (Tables R1,R3): 4-bit TSLD consistently exhibits significantly more robust performance compared to OPTQ and/or AWQ, the latest SOTA PTQ [9][R1] techniques (we used the code published by the authors of corresponding papers). In particular, TSLD achieved the lowest perplexity and the highest accuracy across all the experiments except SST-2, where its accuracy is in-par with the full-precision case. On the other hand, 4-bit PTQ experienced significant accuracy degradation, especially in NLU tasks like CoLA, MRPC, and RTE. These findings demonstrate that TSLD can robustify the performance of 4-bit quantized GLMs for various natural language tasks, while 4-bit PTQ may suffer some catastrophic failures.
> - W2A16 OPT-1.3B on NLU and CSQA (Tables R1,R2,R3): ternary TSLD consistently outperforms the alternative QAT methods for all the cases tested, demonstrating its superior performance in bridging the accuracy gap with the full-precision cases. Interestingly, ternary TSLD even achieved similar or superior accuracy compared to 4-bit PTQ in many tasks (e.g., CoLA, MRPC, SST-2, PIQA, OpenbookQA), highlighting its benefits on both accuracy and memory savings (W4A16-761.9MB -> W2A16-473.2MB).
> (Note: 2-bit PTQ is not successful - e.g., the perplexity of 2-bit OPT-1.3B on Wikitext-2 is degraded from 16.41(FP16) to 35.26(W2A16) even if both AWQ and GPTQ are applied, cf. Table 7 of [R1]. Our observation concurs: see Table R2.)
> - TSLD on GPT-Neo-1.3B and LLaMA-7B (Table R4): We tested the general applicability of TLSD on popular open-sourced GLM models (GPT-Neo-1.3B, LLaMA-7B) using the PTB and GSM8K datasets. Our findings indicate that TSLD consistently surpassed the competitor, Logit KD [20, QuantGPT]. Particularly in the GSM8K dataset for LLaMA-7B, TSLD equaled the perplexity while attaining higher accuracy, demonstrating its capability to mitigate overfitting issues. However, we observed that ternary weight quantization posed greater challenges on LLaMA-7B, a similar phenomenon also noted in [R1] (see Table 8 for PPL increase of 3-bit OPT vs LLaMA). This difficulty seems to correlate with the broader weight range found in LLaMA-7B. As a result, we are exploring methods to tune the diverse weight distributions of GLMs amenable for quantization, as an interesting future research.
> - W4A16 OPT-2.7B on GSM8K (Table R5): We also added the experimental results of various 4-bit weight quantization techniques on OPT-2.7B for GSM8K task, which was initially missed in Table 2 of our paper. Note that TSLD consistently achieved the lowest perplexity and highest accuracy, and all QAT-KD results outperform the accuracy of 4-bit PTQ.
>
> **Comment 2) Experimental results on various data types**
> > Weakness 2) “It would be great for the readers to see the accuracy and perplexity numbers for all data types (int8, int4, and int2).”
> Weakness 4) “Report accuracy and perplexity results for all data types (int8, int4, and int2).”
>
> **Answer 2)** Since the primary focuses of our paper is sub-4-bit QAT, we emphasized ternary and INT4 weight quantization as detailed in Tables 1-3 and R1-R5. Note that INT4 TSLD achieved accuracy and perplexity that matched or surpassed full-precision cases. But we acknowledge the reviewer's suggestion to include results for a complete set of data-types and will incorporate this in the final version.
>
> **Comment 3)**
> > Question 1) ”It's not clear to me whether table 1 shows the training or the test perplexity for the PTB dataset.”
>
> **Answer 3)** Table 1 reported the “evaluation perplexity,” the perplexity measured on the evaluation set of the PTB dataset.

---

> > ### Comment · Reviewer_EGP4 · 2023-08-17
> > **Thanks for addressing my questions**
> >
> > Thanks for addressing my questions and more results on:
> > https://openreview.net/forum?id=FUnEkOkodU&noteId=IArsm06xX2
> >
> > I increase my score to 5

---

> > > ### Author Response · Authors · 2023-08-18
> > >
> > > We deeply appreciate your taking the time to review the additional experimental results and materials we presented in the rebuttal. We will certainly incorporate the valuable comments and discussions provided by the reviewer in the preparation of the revised manuscript.

---

### Author Rebuttal · Authors · 2023-08-10

We sincerely thank the reviewers for their invaluable feedback. To effectively respond to your inquiries, we have attached a PDF containing new experimental results and figures. Detailed responses to each question raised by the reviewers can be found in the respective sections, with references to the data included in this PDF. Once again, we express our gratitude for your time and insights.

---

### Comment · Area_Chair_zfMD · 2023-08-17

Reminder: the author-reviewer discussion is closed on Aug 21st 1pm EDT.

---

### Decision · Program_Chairs · 2023-09-21

**Decision:**

Accept (poster)

**Comment:**

The paper proposes token-scaled logit distillation to prevent overfitting and provide superior learning from the teacher model and ground truth. The reviewers agree to accept the paper. The problem itself is important in practice to better utilize LLMs, so the paper would potentially help other LLM researchers. The comments and some author response can also be taken into consideration for the camera-ready version.